# On permutation-invariant neural networks: A survey

## Abstract

Conventional machine learning algorithms have traditionally been designed under the assumption that input data follows a vector-based format, with an emphasis on vector-centric paradigms. However, as the demand for tasks involving set-based inputs has grown, there has been a paradigm shift in the research community towards addressing these challenges. In recent years, the emergence of neural network architectures such as Deep Sets and Transformers has presented a significant advancement in the treatment of set-based data. These architectures are specifically engineered to naturally accommodate sets as input, enabling more effective representation and processing of set structures. Consequently, there has been a surge of research endeavors dedicated to exploring and harnessing the capabilities of these architectures for various tasks involving the approximation of set functions. This comprehensive survey aims to provide an overview of the diverse problem settings and ongoing research efforts pertaining to neural networks that approximate set functions. By delving into the intricacies of these approaches and elucidating the associated challenges, the survey aims to equip readers with a comprehensive understanding of the field. Through this comprehensive perspective, we hope that researchers can gain valuable insights into the potential applications, inherent limitations, and future directions of set-based neural networks. Indeed, from this survey we gain two insights: i) Deep Sets and its variants can be generalized by differences in the aggregation function, and ii) the behavior of Deep Sets is sensitive to the choice of the aggregation function. From these observations, we show that Deep Sets, one of the well-known permutation-invariant neural networks, can be generalized in the sense of a quasi-arithmetic mean.

## 1 Introduction

In recent years, machine learning has achieved significant success in many fields, and many typical machine learning algorithms handle vectors as their input and output (Zhou, 2021; Islam, 2022; Erickson et al., 2017; Jordan & Mitchell, 2015; Mitchell, 1997). For example, some of these applications include image recognition (Sonka et al., 2014; Minaee et al., 2021; Guo et al., 2016), natural language processing (Strubell et al., 2019; Young et al., 2018; Otter et al., 2020; Deng & Liu, 2018), and recommendation systems (Portugal et al., 2018; Melville & Sindhwani, 2010; Zhang et al., 2019b).

However, with the advancement of the field of machine learning, there has been a growing emphasis on the research of algorithms that handle more complex data structures in recent years. In this paper, we consider machine learning algorithms that deal with sets (Hausdorff, 2021; Levy, 2012; Enderton, 1977) as one such data structure. Here is a couple of examples of set data structures:

- **Set of vector data.** For example a set of image vectors.

- **Point cloud data.** Point cloud data consists of a set of data points, each represented by its spatial coordinates in a multi-dimensional space.

Considering machine learning algorithms that handle sets allows us to leverage the representations of these diverse types of data. Certainly, the goal here is to utilize machine learning models to approximate set

functions. Set functions, in this context, are mathematical functions that operate on sets of elements or data points. These functions capture various properties, relationships, or characteristics within a given set. However, when dealing with complex data and large sets, it can be challenging to directly model or compute these set functions using traditional model architectures. Therefore, we need to consider a specialized model architecture specifically designed for the approximation of set functions.

In particular, a notable characteristic of set functions, when compared to vector functions, is their permutation invariance. Permutation invariance, in the context of set functions or set-based data, means that the output of the function remains the same regardless of the order in which elements of the set are arranged. In other words, if you have a set of data points and apply a permutation (rearrangement) to the elements within the set, a permutation-invariant function will produce the same result. This property is crucial when dealing with sets of data where the order of elements should not affect the function's evaluation. In Section 2 we introduce a more formal definition. Popular conventional neural network architectures like VGG (Simonyan & Zisserman, 2014) and ResNet (He et al., 2016), or more recently proposed ResNeXt (Xie et al., 2017), EfficientNet (Tan & Le, 2019) and ResNest (Zhang et al., 2022b) do not inherently possess the permutation-invariant property. Hence, the primary research interest lies in determining what neural network architectures can be adopted to achieve the permutation-invariant property while maintaining performance and expressive capabilities similar to those achieved by conventional models. In Section 3, we introduce such model architectures and provide the overview of the objective tasks in Section 4. Furthermore, there have been several theoretical analyses of neural network approximations for such permutation-invariant functions, and Section 5 introduces them. To evaluate such research, datasets for performance assessment of approximate set functions are essential. In Section 6, we list some of the well-known datasets for this purpose. Finally, we propose a novel generalization of Deep Sets in Section 7.

Our contributions are summarized as follows.

- We provide a comprehensive survey of permutation-invariant neural network architectures and the tasks/datasets they handle (Section 3, 4 and 6).

- We also highlight existing results of theoretical analysis related to the generalization of Deep Sets in the sense of Janossy pooling (Section 5).

- Finally, based on the above survey results we remark that Deep Sets and its variants can be generalized in the sense of quasi-arithmetic mean. We use this concept to propose a special class of Deep Sets, namely Hölder's Power Deep Sets (Section 7).

## 2 Preliminary and related concepts

First, we introduce the necessary definitions and notation. Let $\mathcal{V} := \{1, \ldots, |\mathcal{V}|\}$ be the ground set.

**Definition 2.1.** Let $\varphi \colon \mathcal{V} \to \mathbb{R}^d$ be the mapping from each element of the ground set $\mathcal{V}$ to the corresponding $d$-dimensional vector with respect to the indices as $\varphi(s_i) = \boldsymbol{s}_i \in \mathbb{R}^d$ for all $s_i \in \mathcal{S} \subseteq \mathcal{V}$. Furthermore, when it is clear from the context, we identify $\mathcal{S} \subseteq \mathcal{V}$ with the set of vectors obtained by this mapping as $\mathcal{S} = \{s_1, \ldots, s_{|\mathcal{S}|}\} = \{\varphi(s_1), \ldots, \varphi(s_{|\mathcal{S}|})\} = \{\boldsymbol{s}_1, \ldots, \boldsymbol{s}_{|\mathcal{S}|}\}$.

For the case of dealing with subsets, we also provide the following definitions.

**Definition 2.2.** Denote the set of all subsets of a set $\mathcal{V}$, known as the power set of $\mathcal{V}$, by $2^{\mathcal{V}}$.

### 2.1 Permutation invariance

There are several key differences between functions that take general vectors as inputs and functions that take sets as inputs.

**Definition 2.3** (Tuple). A tuple, or an ordered $n$-tuple is a set of $n$ elements with an order associated with them. If $n$ elements are represented by $x_1, x_2, \ldots, x_n$, then we write the ordered $n$-tuple as $(x_1, x_2, \to, x_n)$.

The concept of set permutations is important when dealing with set functions. We therefore introduce the following definition of the set of all permutations of any subset.

**Definition 2.4.** Let $\Pi_{\mathcal{S}}$ be the set of all permutations of a tuple $\mathcal{S}$.

Let $\Phi\colon \cup_{k=1}^{|\mathcal{V}|} \mathcal{V}^k \to \mathbb{R}$ be a function defined on tuples.

**Definition 2.5** (Permutation invariant). A function $\Phi\colon \cup_{k=1}^{|\mathcal{V}|} \mathcal{V}^k \to \mathbb{R}$ is said to be permutation invariant if $\Phi(\mathcal{S}) = \Phi(\pi_{\mathcal{S}}\mathcal{S})$ for any tuple $\mathcal{S}$ and its arbitrary permutation $\pi_{\mathcal{S}} \in \Pi_{\mathcal{S}}$.

Here, $\Phi$ can be perceived as a set function if and only if $\Phi$ is permutation invariant. Conversely, a set function $f\colon 2^{\mathcal{V}} \to \mathbb{R}$ induces a pertmutation invariant function. on tuples. On the other hand, function that changes its value depending on the permutation of the input set is called a permutation-sensitive function. Neural networks tasked with approximating set functions face unique challenges and requirements compared to conventional vector-input functions. In order to accurately model and capture the characteristics of sets, these networks need to fulfill the aforementioned properties. First, permutation invariance ensures that the output of the network remains consistent regardless of the order in which elements appear in the set. This is crucial for capturing the inherent structure and compositionality of sets, where the arrangement of elements does not affect the overall meaning or outcome. Second, equivariance to set transformations guarantees that the network's behavior remains consistent under operations such as adding or removing elements from the set. This property ensures that the network can adapt to changes in set size without distorting its output. By satisfying these properties, neural networks can effectively model and approximate set functions, enabling them to tackle a wide range of set-based tasks in various domains.

## 2.2 Permutation equivariance

Along with permutation invariance, another important concept is permutation equivariance. Permutation equivariant is defined as follows.

**Definition 2.6** (Permutation equivariant). Let $\mathcal{U} = \cup_{k=1}^{|\mathcal{V}|} \mathcal{V}^k$. A function $\Phi\colon \mathcal{U} \times \mathcal{U} \to \mathcal{U}$ is said to be permutation equivariant if permutation of the input instances permutes the output labels as

$$f(\{s_{\pi_{\mathcal{S}}(1)}, \ldots, s_{\pi_{\mathcal{S}}(|\mathcal{S}|)}\}) = \{f_{\pi_{\mathcal{S}}(1)}(\mathcal{S}), \ldots, f_{\pi_{\mathcal{S}}(|\mathcal{S}|)}(\mathcal{S})\}. \tag{1}$$

This equivariance property is important in the supervised setting (Zaheer et al., 2017). In recent studies, it has been theoretically shown that permutation invariant transformations can be constructed by combining multiple permutation equivariance transformations (Fei et al., 2022).

## 3 Model architectures for approximating set functions

In this section, we give an overview of neural network architectures approximating set functions. Table 1 summarizes architectures as well as the corresponding applications.

In particular, we focus on architectures following the idea of Deep Sets, which demonstrated universality results for permutation-invariant inputs. However, prior to that, several similar studies on related architectures also exist (Gens & Domingos, 2014; Cohen & Welling, 2016). For example, invariance can be achieved by pose normalization using an equivariant detector (Lowe, 2004; Jaderberg et al., 2015), or by averaging a possibly nonlinear function over a group (Reisert, 2008; Manay et al., 2006; Kondor, 2007).

### 3.1 Deep Sets

One seminal work for approximating set functions by neural networks is Deep Sets (Zaheer et al., 2017). The framework of Deep Sets is written as

$$f(\mathcal{S}) := \rho\left(\sum_{\boldsymbol{s} \in \mathcal{S}} \phi(\boldsymbol{s})\right), \tag{2}$$

| Architecture | Novelties and contributions | Applied tasks |
|---|---|---|
| Deep Sets (Zaheer et al., 2017) | The universality result for permutation-invariance and sum-decomposability. | General set function approximation
Point cloud classification
Set expansion
Set retrieval
Image tagging
Set anomaly detection |
| PointNet (Qi et al., 2017a) | The max-decomposition architecture. | Point cloud classification
Point cloud segmentation |
| PointNet++ (Qi et al., 2017b) | | |
| SetNet (Zhong et al., 2018) | Utilizing NetVLAD layer for the set retrieval task. | Set retrieval |
| Set Transformer (Lee et al., 2019) | Utilizing Transformer architecture for permutation-invariant inputs. | General set function approximation
Point cloud classification
Set anomaly detection |
| DSPN (Zhang et al., 2019b) | A model that predicts a set of vectors from another vector. | Set reconstruction
Bounding box prediction |
| iDSPN (Zhang et al., 2021) | The novel concept of exclusive multiset-equivariance. | Class specific numbering
Random multisets reconstruction
Object property prediction |
| SetVAE (Kim et al., 2021b) | The VAE-based set generation model. | Set generation |
| Slot Attention (Locatello et al., 2020) | A new variant of permutation-invariant attention mechanism. | Object discovery
Set prediction |
| Deep Sets++ & Set Transformer++ (Zhang et al., 2022c) | The Set Normalization as the alternative normalization layer. | General set function approximation
Point cloud classification
Set anomaly detection |
| PointCLIP (Zhang et al., 2022d) | CLIP for permutation-invariant inputs. | Point cloud classification |
| Perceiver (Jaegle et al., 2021b) | Introduction of the computationally efficient learnable query in the framework of Set Transformer. | General set function approximation
Point cloud classification |
| Perceiver IO (Jaegle et al., 2021a) | Perceiver for multimodal outputs. | General set function approximation
Multimodal set embedding |
| Perceiver VL (Tang et al., 2023) | Perceiver-based architecture for vision and language tasks. | Set retrieval |

Table 1: Neural network architectures for approximating set functions.

for set $\mathcal{S}$, and two functions $\phi$, $\rho$. We can see that Deep Sets architecture satisfies the permutation-invariant 2.5. Moreover it can represent the set function by using arbitrary neural networks $\phi$ and $\rho$. It is also known that Deep Sets have the universality for permutation-invariant and sum-decomposability (see Section 5.1 for more details).

One fundamental property for approximating set functions is the sum-decomposability.

**Definition 3.1.** The function $f$ is said to be sum-decomposable if there are functions $\rho$ and $\phi$ such that

$$f(\mathcal{S}) = \rho \left( \sum_{s \in \mathcal{S}} \phi(s) \right) \tag{3}$$

for any $\mathcal{S} \subseteq \mathcal{V}$. In this case, we say that $(\rho, \phi)$ is a sum-decomposition of $f$. Given a sum-decomposition $(\rho, \phi)$, we write $\Phi(\mathcal{S}) := \sum_{s \in \mathcal{S}} \phi(s)$. With this notation, we can write Eq. 3 as $f(\mathcal{S}) = \rho(\Phi(\mathcal{S}))$. We may also refer to the function $\rho \circ \Phi$ as a sum-decomposition.

Then, we can see that the architecture of Deep Sets satisfies the sum-decomposability.

## 3.2 PointNet and PointNet++

The most well-known architecture developed for processing point cloud data is PointNet (Qi et al., 2017a). One of the important differences between PointNet and Deep Sets is their pooling operation. For instance, PointNet employs global max pooling, while global sum pooling is adopted in Deep Sets. This implies that we can write PointNet architecture as follows:

$$f(\mathcal{S}) := \rho \left( \max_{s \in \mathcal{S}} \phi(s) \right). \tag{4}$$

Therefore, we can express the architectures of Deep Sets and PointNet in a unified notation.

A function that can be written in the form of Eq. 2 is referred to as sum-decomposable, and the function that can be written in the form of Eq. 4 is called max-decomposable.

The universality result of sum-decomposability in Deep Sets suggests that a similar result holds for max-decomposable functions, as indicated in subsequent studies (Wagstaff et al., 2022).

## 3.3 Set Transformer

In recent years, the effectiveness of Transformer architectures (Vaswani et al., 2017; Kimura & Tanaka, 2019; Lin et al., 2022) has been reported in various tasks that neural networks tackle, such as natural language processing (Kalyan et al., 2021; Wolf et al., 2019; Kitaev et al., 2020; Beltagy et al., 2020), computer vision (Han et al., 2022b; Khan et al., 2022; Dosovitskiy et al., 2020; Arnab et al., 2021; Zhou et al., 2021a), and time series analysis (Wen et al., 2022; Zhou et al., 2021b; Zerveas et al., 2021). Set Transformer (Lee et al., 2019) utilize the Transformer architecture to handle permutation-invariant inputs. Similar architectures have also been proposed for point cloud data (Guo et al., 2021; Park et al., 2022; Zhang et al., 2022a; Liu et al., 2023).

The architectures of Deep Sets and PointNet, as evident from Eq. 2 and 4, operate by independently transforming each element of the input set and then aggregating them. However, this approach neglects the relationships between elements, leading to a limitation. On the other hand, the Set Transformer addresses this limitation by introducing an attention mechanism, which takes into account the relationships between two elements in the input set. We can write this operation as follows.

$$f(\mathcal{S}) = \rho \left( \frac{1}{\tau(|\mathcal{S}|, 2)} \sum_{\mathcal{T} \in \mathcal{S}_{(2)}} \phi(t_1, t_2) \right), \quad (t_1, t_2 \in \mathcal{T}) \tag{5}$$

where $\tau(|\mathcal{S}|, 2) = \frac{|\mathcal{S}|}{(|\mathcal{S}|-2)!}$. Eq. 5 can be viewed as performing permutation-invariant operations on 2-subsets of permutations of the input set. Indeed, $\phi(t_1, t_2)$ is usually split up in attention weights $w(t_1, t_2)$ and values $v(t_2)$ and a softmax acts on the weights.

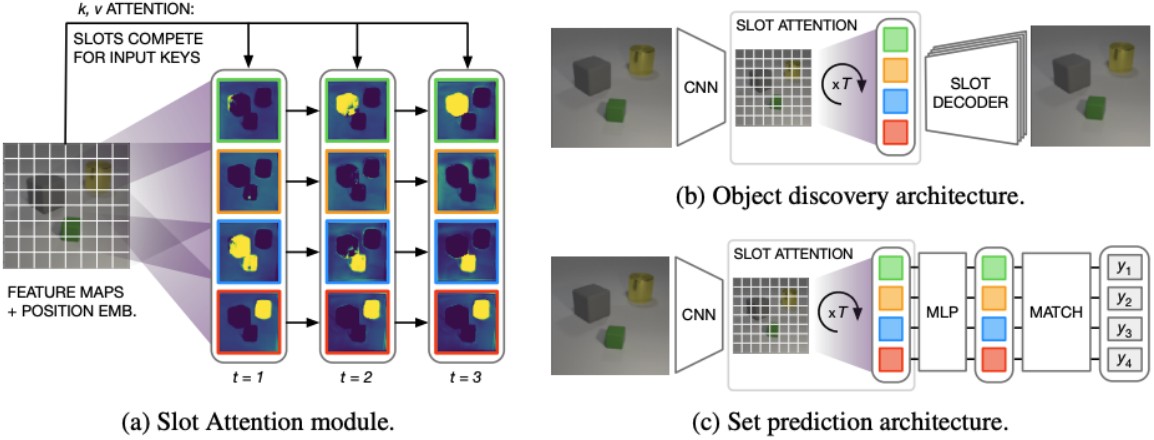

Figure 1: Slot Attention module and example applications to unsupervised object discovery and supervised set prediction with labeled targets, from Figure 1 of Locatello et al. (2020). This figure is cited to illustrate the behavior of the Slot Attention module.

As introduced in Section 5.1, recent research has revealed that Deep Sets, PointNet, Set Transformer, and their variants can be regarded as special cases of a function class called Janossy Pooling (Murphy et al., 2018). Moreover, there have been several discussions regarding the generalization of Transformer and Deep Sets (Kim et al., 2021a; Maron et al., 2018).

### 3.4 Deep Sets++ and Set Transformer++

Many neural network architectures include normalization layers such as Layer Norm (Ba et al., 2016), BatchNorm (Ioffe & Szegedy, 2015; Bjorck et al., 2018) or others (Wu & He, 2018; Salimans & Kingma, 2016; Huang & Belongie, 2017). SetNorm (Zhang et al., 2022c) is used for neural networks that take sets as input, based on the result that the normalization layer is permutation-invariant only when the transformation part of the normalization layer deforms all the features with different scales and biases for each feature. Specifically, applying SetNorm to Deep Sets and Set Transformer, referred to as Deep Sets++ and Set Transformer++ respectively, has been shown experimentally to achieve improvements of original Deep Sets and Set Transformer. Furthermore, Zhang et al. (2022c) also releases a dataset called Flow-RBC, which comprises sets of measurement results of red blood cells of patients, aimed at predicting anemia.

### 3.5 DSPN and iDSPN

Deep Set Prediction Networks (DSPN) (Zhang et al., 2019b) propose a model for predicting a set of vectors from another set of vectors. The proposed decoder architecture leverages the fact that the gradients of the set functions with respect to the set are permutation-invariant, and it is effective for tasks such as predicting a set of bounding boxes for a single input image. Also, iDSPN (Zhang et al., 2021) introduced the concept of exclusive multiset-equivariance to allow for the arbitrary ordering of output elements with respect to duplicate elements in the input set. They also demonstrated that by constructing the encoder of DSPN using Fspool (Zhang et al., 2019c), the final output for the input set satisfies exclusive multiset-equivariance.

### 3.6 SetVAE

SetVAE (Kim et al., 2021b) is the set generation model based on Variational Auto-Encoder (VAE) (Kingma & Welling, 2013) that takes into account exchangeability, variable-size sets, interactions between elements, and hierarchy. Here, hierarchy refers to the relationships between subsets within a set, such as the hierarchical structure of elements in the set. The concept of the Hierarchical VAE was introduced in the context of high-resolution image generation (Sønderby et al., 2016; Vahdat & Kautz, 2020), and this study utilizes it

for set generation. As an application of the idea of SetVAE, SCHA-VAE (Giannone & Winther, 2022) for generating a few shot image is also proposed.

### 3.7 PointCLIP

One of the learning strategies that has garnered significant attention in the field of machine learning in recent years is Contrastive Language-Image Pre-training (CLIP) (Radford et al., 2021; Shen et al., 2021; Luo et al., 2022). PointCLIP (Zhang et al., 2022d) adopts CLIP for permutation-invariant neural networks. PointCLIP encodes point cloud data using CLIP and achieves category classification for 3D data by examining their positional relationships with category texts. Furthermore, PointCLIPv2 (Zhu et al., 2023) is an improvement of PointCLIP, achieved through a dialogue system.

### 3.8 Slot Attention

The Slot Attention (Locatello et al., 2020) mechanism was proposed to obtain representations of arbitrary objects in images or videos in an unsupervised manner. Slot Attention employs an iterative attention mechanism to establish a mapping from its inputs to the slots (see Fig. 1). The slots are initially set at random and then refined at each iteration to associate with specific parts or groups of the input features. The process involves randomly sampling initial slot representations from a common probability distribution. It is proven that Slot Attention is

  i) permutation invariance with respect to the input;

  ii) permutation equivariance with respect to the order of the slots.

Zhang et al. (2022e) pointed out two issues with slot attention: the problem of single objects being bound to multiple slots (soft assignments) and the problem of multiple slots processing similar inputs, resulting in multiple slots having averaged information about the properties of a single object (Lack of tiebreaking). To address these issues, they leverage the observation that part of the slot attention processing can be seen as one step of the Sinkhorn algorithm (Sinkhorn, 1964), and they propose a method to construct slot attention to be exclusive multiset-equivariant without sacrificing computational efficiency.

Chang et al. (2022) argue that slot attention suffers from the issue of unstable backward gradient computation because it performs sequential slot updates during the forward pass. Specifically, as training progresses, the spectral norm of the model increases. Therefore, they experimentally demonstrated that by replacing the iterative slot updates with implicit function differentiation at the fixed points, they can achieve stable backward computation without the need for ad hoc learning stabilization techniques, including gradient clipping (Pascanu et al., 2013; Zhang et al., 2019a), learning rate warmup (Goyal et al., 2017; Liu et al., 2019) or adjustment of the number of iterative slot updates. Kipf et al. (2021) point out that initializing slots through random sampling from learnable Gaussian distributions during the sequential updating process may lead to instability in behavior. Based on this, Jia et al. (2022) propose stabilizing the behavior by initializing slots with fixed learnable queries, namely BO-QSA. Vikström & Ilin (2022) propose the ViT architecture and a corresponding loss function within the framework of Masked Auto Encoder to acquire object-centric representations. Furthermore, DINOSAUR (Seitzer et al., 2022) learns object-centric representations based on higher-level semantics by optimizing the reconstruction loss with ViT features. Slotformer (Wu et al., 2022) is proposed as a transformer architecture that predicts slots autoregressively, and it has been reported to achieve high performance. There are many other variants of slot attention along with their respective applications, such as SIMONe (Kabra et al., 2021), EfficientMORL (Emami et al., 2021), OSRT (Sajjadi et al., 2022), OCLOC (Yuan et al., 2022), SAVi (Kipf et al., 2021) or SAVi++ (Elsayed et al., 2022).

### 3.9 Perceiver

In the attention mechanism, increasing the length of the sequence is an important issue to be addressed. Perceiver (Jaegle et al., 2021b) tackles this problem by introducing computationally efficient learnable queries.

**Taxonomy of approximating set functions**

Figure 2: Taxonomy of approximating set functions. Several tasks can be considered as special cases of other tasks. For example, set retrieval, which is a set version of image retrieval, can be regarded as a kind of subset selection that extracts a subset from a set.

Although this framework can guarantee computational efficiency with the expressive power like Set Transformer, a limitation is that it is only applicable to unimodal classification tasks. Perceiver IO (Jaegle et al., 2021a) handles multimodal output by adding the decoder to Perceiver. Perceiver VL (Tang et al., 2023) also allows Perceiver-based architectures to be applied to vision and language tasks. In addition, Framingo (Alayrac et al., 2022), one of the foundation models for various vision and language downstream tasks, employs the iterative latent cross-attention proposed by Perceiver in its architecture.

## 4 Tasks of approximating set functions

In this section, we organize the tasks addressed by neural networks that approximate set functions. Figure 2 shows the taxonomy of approximating set functions.

### 4.1 Point cloud processing

Deep Sets, PointNet, and Set Transformer can be generalized in terms of the differences in the aggregation operations of elements within a set. However, specific aggregation operations are also proposed when the input set consists of point clouds. CurveNet (Xiang et al., 2021) proposes to treat point clouds as undirected graphs and represent curves as walks within the graph, thereby aggregating the points.

### 4.2 Set retrieval and subset selection

There exists a set retrieval task that generalizes the image retrieval task (Datta et al., 2008; Smeulders et al., 2000; Rui et al., 1999) to sets. The goal of the set retrieval system is to search and retrieve sets from the large pool of sets (Zhong et al., 2018).

**Subset selection**   Subset selection is the task of selecting a subset of elements from a given set in a way that retains some meaningful criteria or properties. Ou et al. (2022) introduced the low-cost annotation method for subset selection and demonstrated its effectiveness.

SetNet (Zhong et al., 2018) is an architecture designed for set retrieval, which uses NetVLAD layer (Jin et al., 2021) instead of conventional pooling layers. In this paper, the Celebrity Together dataset is proposed specifically for set retrieval.

### 4.3   Set generation and prediction

Methods for set prediction can be broadly categorized into the following two approaches:

- distribution matching: approximates $P(\mathcal{Y}|\boldsymbol{x})$ for a set $\mathcal{Y}$ and an input vector $\boldsymbol{x}$;

- minimum assignment: calculates loss function between the assigned pairs.

**Distribution matching**   Deep Set Prediction Networks (DSPN) (Zhang et al., 2019b) propose a model for predicting a set of vectors from another set of vectors. The proposed decoder architecture leverages the fact that the gradients of the set functions with respect to the set are permutation-invariant, and it is effective for tasks such as predicting a set of bounding boxes for a single input image. Also, iDSPN (Zhang et al., 2021) introduced the concept of exclusive multiset-equivariance to allow for the arbitrary ordering of output elements with respect to duplicate elements in the input set. PointGlow (Sun et al., 2020) applies the flow-based generative model for point cloud generation. With similar ideas, Biloš & Günnemann (2021) propose a continuous normalizing flow for sequential invariant vector data, and Zwartsenberg et al. (2023) propose Conditional Permutation Invariant Flows, which extend it to allow conditional generation.

**Minimum assignment**   In the minimum assignment approach, there is freedom in choosing the distance function, but the latent set prediction (LSP) framework (Preechakul et al., 2021) relaxes this and provides convergence analysis. SetVAE (Kim et al., 2021b) is the set generation model based on VAE (Kingma & Welling, 2013) that takes into account exchangeability, variable-size sets, interactions between elements, and hierarchy.

Carion et al. (2020) consider object detection as a bounding box set prediction problem and propose an assignment-based method using a transformer, called Detection Transformer (DETR). Inspired by this identification of object detection and set prediction, many studies have been conducted using similar strategies (Hess et al., 2022; Carion et al., 2020; Ge et al., 2021; Misra et al., 2021). It has been reported that DETR can achieve SOTA performance, but its long training time is known to be a bottleneck. Sun et al. (2021) reconsidered the difficulty of DETR optimization and pointed out two causes of slow convergence: Hungarian loss and Transformer cross-attention mechanism. They also proposed several methods to solve these problems and showed their effectiveness through experiments.

Zhang et al. (2020) pointed out that methods optimizing assignment-based set loss inadvertently restrict the learnable probability distribution during the loss function selection phase and assume implicitly that the generated target follows a unimodal distribution on the set space, and proposed techniques to address these limitations.

### 4.4   Set matching

The task of estimating the degree of matching between two sets is referred to as set matching.

Set matching can be categorized into two cases: the homogeneous case and the heterogeneous case. In the homogeneous case, both input sets consist of elements from the same category or type. On the other hand, in the heterogeneous case, the input sets contain elements from different categories or types. Saito et al. (2020) proposes a novel approach for the heterogeneous case, which has not been addressed before. To solve set matching problems, it often relies on learning with negative sampling, and Kimura (2022) provides theoretical analyses for such problem settings. Furthermore, recent research has also reported the task-specific distribution shift in set matching tasks (Kimura, 2023).

### 4.5 Neural processes

The Neural Process family (Garnelo et al., 2018b;a; Jha et al., 2022), which is an approximation of probabilistic processes using neural networks, has been studied extensively. Garnelo et al. (2018a) first introduced the idea of conditional Neural Processes which model the conditional predictive distribution $p(f(T)|T, C)$, where $C$ is the labeled dataset and $T$ is the unlabeled dataset. One of the necessary conditions for defining a probabilistic process is permutation invariance. In the case of CNPs, to fulfill this condition, the encoder-decoder parts employ the architecture of Deep Sets. However, the output of CNPs consists of a pair of prediction mean and standard deviation, and it is not possible to sample functions like in actual probabilistic processes. Neural Processes (NPs) (Garnelo et al., 2018b) enable function sampling by introducing latent variables into the framework of CNPs.

Kim et al. (2018) showed that NPs are prone to underfitting. They argued that this problem can be solved by introducing an attention mechanism, and proposed Attentive Neural Processes.

### 4.6 Approximating submodular functions

Submodular set function (Fujishige, 2005; Lovász, 1983; Krause & Golovin, 2014) is one of the important classes of set functions, and there exist many applications. First, we introduce the definitions and known results for the submodular set function.

**Definition 4.1.** A function $f\colon 2^{\mathcal{V}} \to \mathbb{R}$ is called submodular if it satisfies

$$f(\mathcal{S}) + f(\mathcal{T}) \geq f(\mathcal{S} \cup \mathcal{T}) + f(\mathcal{S} \cap \mathcal{T}), \tag{6}$$

for any $\mathcal{S}, \mathcal{T} \subseteq \mathcal{V}$.

**Definition 4.2.** A function $f\colon 2^{\mathcal{V}} \to \mathbb{R}$ is supermodular if $-f$ is submodular.

**Definition 4.3.** A function that is both submodular and supermodular is called modular.

If $f\colon 2^{\mathcal{V}} \to \mathbb{R}$ is a modular function, we have

$$f(\mathcal{S}) + f(\mathcal{T}) = f(\mathcal{S} \cap \mathcal{T}) + f(\mathcal{S} \cup \mathcal{T}), \tag{7}$$

for any $\mathcal{S}, \mathcal{T} \subseteq \mathcal{V}$.

**Proposition 4.1** ((Fujishige, 2005)). If $f\colon 2^{\mathcal{V}} \to \mathbb{R}$ is modular, it may be written as

$$f(\mathcal{S}) = f(\emptyset) + \sum_{\boldsymbol{s} \in \mathcal{S}} (f(\{\boldsymbol{s}\}) - f(\emptyset)) \tag{8}$$

$$= c + \sum_{\boldsymbol{s} \in \mathcal{S}} \phi(\boldsymbol{s}), \tag{9}$$

for some $\phi\colon \mathcal{V} \to \mathbb{R}$.

From the above definitions and results, we have the following proposition for permutation-invariant neural networks.

**Proposition 4.2.** We assume that $\rho(\boldsymbol{s}) = \boldsymbol{s}$ for $\boldsymbol{s} \in \mathcal{S}$. Then, for permutation-invariant neural networks, we have

   i) The architecture of Deep Sets takes the form of the modular function;

   ii) The architecture of PointNet takes the form of the submodular function,

for all $\boldsymbol{x} \in \mathcal{V}$.

*Proof.* i) Let $c = 0$ and $\rho(\boldsymbol{s}) = \boldsymbol{s}$ in Eq. 9, we can confirm the statement.

ii) For $\mathcal{S}, \mathcal{T} \subseteq \mathcal{V}$, let

$$a := \max_{\boldsymbol{s} \in \mathcal{S}} \phi(\boldsymbol{s}), \quad b := \max_{\boldsymbol{s} \in \mathcal{T}} \phi(\boldsymbol{s}), \quad c := \max_{\boldsymbol{s} \in \mathcal{S} \cap \mathcal{T}} \phi(\boldsymbol{s}), \quad d := \max_{\boldsymbol{s} \in \mathcal{S} \cup \mathcal{T}} \phi(\boldsymbol{s}).$$

Then we need to show

$$a + b \geq c + d.$$

Here, we can see that $a, b \geq c$ and thus $\min(a, b) \geq c$. Similarly, we have $\max(a, b) = d$, and

$$a + b = \min(a, b) + \max(a, b)$$
$$\geq c + d.$$

Then, we have the proof. $\qquad\qquad\qquad\qquad\qquad\qquad\qquad\qquad\qquad\qquad\qquad\qquad\square$

Deep Submodular Functions (Dolhansky & Bilmes, 2016) is one of the seminal works on learning-based submodular functions. Furthermore, Tschiatschek et al. (2016) proposes a probability model where the energy function is represented by a parametric submodular function.

### 4.7 Person re-identification

Person re-identification (Zheng et al., 2015; Liao et al., 2015; Zheng et al., 2017) can be viewed as an approximation problem of set functions since it involves selecting the target element from a set of person images as input. The HAP2S loss (Yu et al., 2018) is proposed with the aim of efficient point-to-set metric learning for the Person re-identification task.

As a variant task of person re-identification, there is also group re-identification (Wei-Shi et al., 2009; Zheng et al., 2014; Lisanti et al., 2017), which involves identifying groups of individuals in images or videos. Lisanti et al. (2017) introduced a visual descriptor that achieves invariance both to the number of subjects and to their displacement within the image in this task. Xiong & Lai (2023) addresses GroupReID between RGB and IR images.

### 4.8 Other tasks

**Metric learning** In traditional video-based action recognition methods, task recognition often involves extracting subtasks and performing temporal alignment. However, Wang et al. (2022) suggests that, in some cases, the order of subtasks may not be crucial, and there could be alternative approaches that can achieve similar results. To perform distance learning between the query video and the contrastive videos, a set matching metric is introduced (Wang et al., 2022).

Sinha & Fleuret (2023) propose the permutation-invariant transformer-based model that can estimate the Earth Mover's Distance in quadratic order with respect to the number of elements. They report the effectiveness of the Sinkhorn algorithm in cases where there are constraints on computational costs, as increasing the number of iterations in the Sinkhorn algorithm improves accuracy compared to their proposed algorithm. Cuturi (2013) proposes a parallelizable Sinkhorn algorithm operating on multiple pairs of histograms that function within the GPU environment.

**XAI** Explainable Artificial Intelligence (XAI) aims to explain the behavior of machine learning models (Gunning et al., 2019; Tjoa & Guan, 2020; Kimura & Tanaka, 2020). Several studies are exploring the combination of approximating set functions and XAI techniques. Cotter et al. (2018) and Cotter et al. (2019) introduce an architecture for approximating interpretable set functions that maintain performance comparable to Deep Sets.

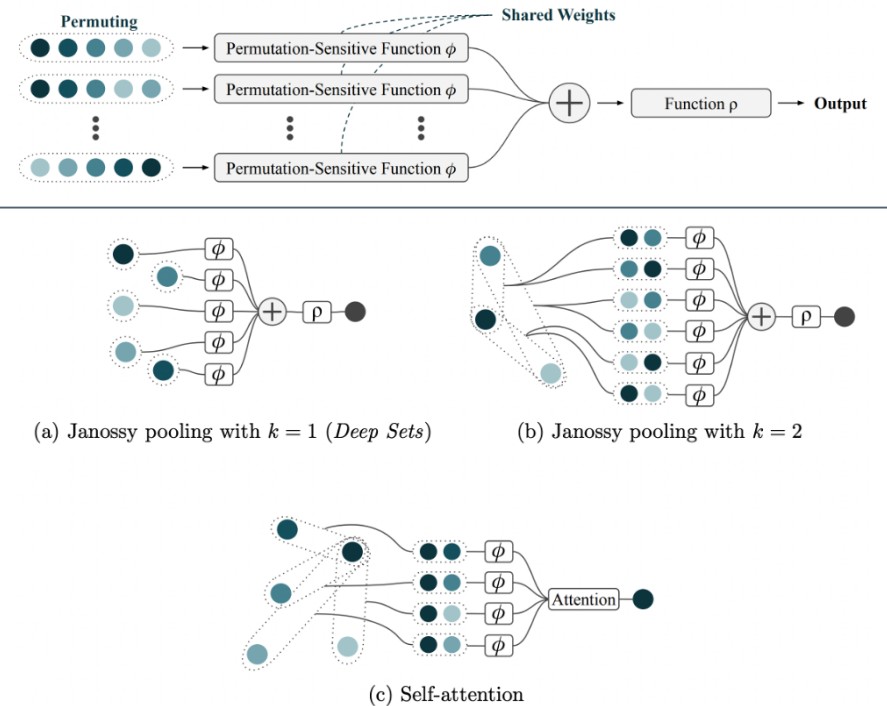

Figure 3: Top panel: The Janossy pooling framework with the same permutation-sensitive network to each possible permutation of the input set, from Figure 1 of Wagstaff et al. (2022). Bottom panel: Different versions and variants of Janossy pooling, from Figure 2 of Wagstaff et al. (2022). These figures are cited to highlight that Janossy pooling is a generalization of Deep Sets and self-attention.

**Federated learning** Federated learning (Kairouz et al., 2021; Li et al., 2021; Konečný et al., 2016) is a framework of machine learning in which data is distributed without aggregation, and its use is expected to improve the efficiency of data processing. Federated Learning is motivated by a general interest in issues such as data privacy, data minimization, and data access rights. Amosy et al. (2024) proposed the application of Deep Sets to federated learning and reported its usefulness.

## 5 Theoretical analysis of approximating set functions

In this section we review several known theoretical results on the approximation of set functions.

### 5.1 Sum-decomposability and Janossy pooling

**Definition 5.1.** Let $(\rho, \phi)$ be a sum-decomposition. Write $\mathcal{Z}$ for the domain of $\rho$ (which is also the codomain of $\phi$, and the space in which the summation happens in Eq. 3. We refer to $\mathcal{Z}$ as the latent space of the sum-decomposition $(\rho, \phi)$.

**Definition 5.2.** Given a space $\mathcal{Z}$, we say that $f$ is sum-decomposable via $\mathcal{Z}$ if $f$ has a sum-decomposition whose latent space is $\mathcal{Z}$.

**Definition 5.3.** The function $f$ is said to be continuously sum-decomposable when there exists a sum-decomposition $(\rho, \phi)$ of $f$ such that both $\rho$ and $\phi$ are continuous. $(\phi, \rho)$ is then a continuous sum-decomposition of $f$.

The objective is to construct models that can effectively represent diverse functions while ensuring that the output remains unchanged when the input elements are permuted. This delicate equilibrium guarantees that

the models can capture the inherent complexities of the problem at hand while upholding the crucial aspect of permutation invariance.

One unifying framework of methods that learns either strictly permutation-invariant functions or suitable approximations is Janossy pooling (Murphy et al., 2018). Janossy pooling is renowned for its remarkable expressiveness, and its universality can be readily illustrated. It is capable of representing any permutation-invariant function, making it a highly versatile framework. This exceptional property highlights the ability of Janossy pooling to capture intricate relationships and patterns within sets. With its flexibility and effectiveness, Janossy pooling serves as a valuable tool for modeling and analyzing permutation-invariant functions, offering a broad spectrum of applications across diverse domains.

**Definition 5.4** (Janossy pooling (Murphy et al., 2018)). For any set $\mathcal{S} \subseteq \mathcal{V}$ and its permutations $\Pi_{\mathcal{S}}$, Janossy pooling is defined as the aggregation of outputs of the permutation-sensitive function $\Phi(\mathcal{S})$ for all possible permutations:

$$\hat{f}(\mathcal{S}) = \frac{1}{|\Pi_{\mathcal{S}}|} \sum_{\pi_{\mathcal{S}} \in \Pi_{\mathcal{S}}} \Phi(\pi_{\mathcal{S}}(\mathcal{S})). \tag{10}$$

We can also consider the post-process $\rho$ as

$$f(\mathcal{S}) = \rho\left(\hat{f}(\mathcal{S})\right), \tag{11}$$

and this is the form of sum-decomposable 3.1, and permutation-invariant 2.5.

It is obvious that the computational complexity of Janossy pooling scales at least linearly in the size of $\Pi_{\mathcal{S}}$, which is $|\mathcal{S}|!$. To address this problem, the following strategies are discussed (Murphy et al., 2018):

  i) sorting: considering only a single canonical permutation, which is obtained by sorting the inputs;

  ii) sampling: aggregating over a randomly-sampled subset of permutations;

  iii) restricting permutation to $k$-subsets: for some $k < |\mathcal{S}|$, let $\mathcal{S}_{\{k\}}$ denote the set of all $k$-subsets from $\mathcal{S}$, and

$$\hat{f}(\mathcal{S}) = \frac{1}{\tau(|\mathcal{S}|, k)} \sum_{\mathcal{T} \in \mathcal{S}_{\{k\}}} \Phi(\mathcal{T}), \tag{12}$$

  where $\tau(|\mathcal{S}|, k) = \frac{|\mathcal{S}|}{(|\mathcal{S}|-k)!}$.

The computational complexity of Eq. 12 is $\mathcal{O}(|\mathcal{S}|^k)$, and for sufficiently small $k$ this gives far fewer that $|\mathcal{S}|!$. Note that the third strategy is the generalization of many practical models (Zaheer et al., 2017; Qi et al., 2017a;b; Lee et al., 2019). Indeed, the case of $k = 1$ is equivalent to Deep Sets Zaheer et al. (2017), and many other current neural network architectures resemble the case of $k = 2$ (see Fig. 3). The function $\hat{f}(\mathcal{S})$ in Eq. 12 is called $k$-ary Janossy pooling.

**Theorem 5.1.** Let $f \colon \mathbb{R}^M \to \mathbb{R}$ be continuous and permutation invariant. Then $f$ has a continuous $k$-ary Janossy representation via $\mathbb{R}^M$ for any choice of $k$.

**Theorem 5.2.** Let $f \colon \mathbb{R}^M \to \mathbb{R}$ be continuous and permutation invariant. Then $f$ has a continuous $M$-ary Janossy representation via $\mathbb{R}$.

The proof of the above theorem is given by examining the definition of Janossy representations.

## 5.2 Expressive power of Deep Sets and PointNet

Recall that the network architectures of PointNet $f_{\text{PN}}$ and Deep Sets $f_{\text{DS}}$ are given as

$$f_{\text{PN}} = \rho \left( \max_{\boldsymbol{s} \in \mathcal{S}} \phi(\boldsymbol{s}) \right), \tag{13}$$

$$f_{\text{DS}} = \rho \left( \sum_{\boldsymbol{s} \in \mathcal{S}} \phi(\boldsymbol{s}) \right). \tag{14}$$

In addition, we can consider the normalized version of Deep Sets $f_{\text{N-DS}}$ as

$$f_{\text{N-DS}} = \rho \left( \frac{1}{|\mathcal{S}|} \sum_{\boldsymbol{s} \in \mathcal{S}} \phi(\boldsymbol{s}) \right). \tag{15}$$

Bueno & Hylton (2021) provides the comparison of representation power and universal approximation theorems of PointNet and Deep Sets. Briefly,

- PointNet (normalized Deep Sets) has the capability to uniformly approximate functions that exhibit uniform continuity in relation to the Hausdorff (Wasserstein) metric.

- When input sets are allowed to be of arbitrary size, only constant functions can be uniformly approximated by both PointNet and normalized Deep Sets simultaneously.

- Even when the cardinality is fixed to a size of $k$, there exists a significant disparity in the approximation capabilities. Specifically, PointNet is unable to uniformly approximate averages of continuous functions over sets, such as the center-of-mass or higher moments, for $k \geq 3$. Furthermore, an explicit lower bound on the error for the learnability of these functions by PointNet is established.

In the following, we give the reproduction of the key statements of Deep Sets (Zaheer et al., 2017).

**Theorem 5.3.** The function $f : 2^{\mathcal{V}} \to \mathbb{R}$ is sum-decomposable if and only if $f$ is a set function.

*Proof.* Let $\Phi(\mathcal{S}) \coloneqq \sum_{\boldsymbol{s} \in \mathcal{S}} \phi(\boldsymbol{s})$. Since $\mathcal{V}$ is countable, each $s \in \mathcal{V}$ can be mapped to a unique element in $\mathbb{N}$ by a bijective function $c \colon \mathcal{V} \to \mathbb{N}$. If we can choose $\phi$ so that $\Phi$ is invertible, then we can write $\rho = f \circ \Phi^{-1}$, and $f = \rho \circ \Phi$. Then $f$ is sum-decomposable via $\mathbb{R}$. $\square$

**Theorem 5.4.** Let $M \in \mathbb{N}$, and $f \colon [0, 1]^M \to \mathbb{R}$ be a continuous permutation-invariant function. Then $f$ is continuously sum-decomposable via $\mathbb{R}^{M+1}$.

**Theorem 5.5.** Deep Sets can represent any continuous permutation-invariant function of $M$ elements if the dimension of the latent space is at least $M + 1$.

In addition, later work (Han et al., 2022a) gives explicit bounds on the number of parameters with respect to the dimension and the target accuracy $\epsilon$. In the following, a set function $f \colon 2^{\mathcal{V}} \to \mathbb{R}$ said to be permutation-invariant and differentiable if it satisfies Def. 2.5 and $f(\varphi(s_1), \ldots, \varphi(s_{|\mathcal{S}|}))$ is differentiable with mapping $\varphi : \mathcal{V} \to \mathbb{R}^d$.

**Theorem 5.6** (Han et al. (2022a)). Let $f \colon 2^{\mathcal{V}} \to \mathbb{R}$ be a continuously-differentiable, permutation-invariant function. Let $0 < \epsilon < \|\nabla f\|_2 \sqrt{|\mathcal{S}|d} \mathcal{S}^{-\frac{1}{d}}$, for any $\mathcal{S} \subseteq \mathcal{V}$ with the mapping $\varphi \colon [1, |\mathcal{V}|] \to \mathbb{R}^d$, where $\|\nabla f\|_2 = \max_{\mathcal{S}} \|f(\mathcal{S})\|_2$. Then, there exists $\phi \colon \mathbb{R}^d \to \mathbb{R}^M$, $\rho \colon \mathbb{R}^M \to \mathbb{R}$, such that

$$\left| f(\mathcal{S}) - \rho \left( \sum_{\boldsymbol{s} \in \mathcal{S}} \phi(\boldsymbol{s}) \right) \right| = \left| f(\mathcal{S}) - \rho \left( \sum_{i=1}^{|\mathcal{S}|} \phi(\varphi(s_i)) \right) \right| < \epsilon, \tag{16}$$

where $M$, the number of feature variables, satisfies the bound

$$M \leq \frac{2^N (\|\nabla f\|_2^2 |\mathcal{S}| d)^{|\mathcal{S}|d/2}}{\epsilon^{|\mathcal{S}|d} |\mathcal{S}|!}. \tag{17}$$

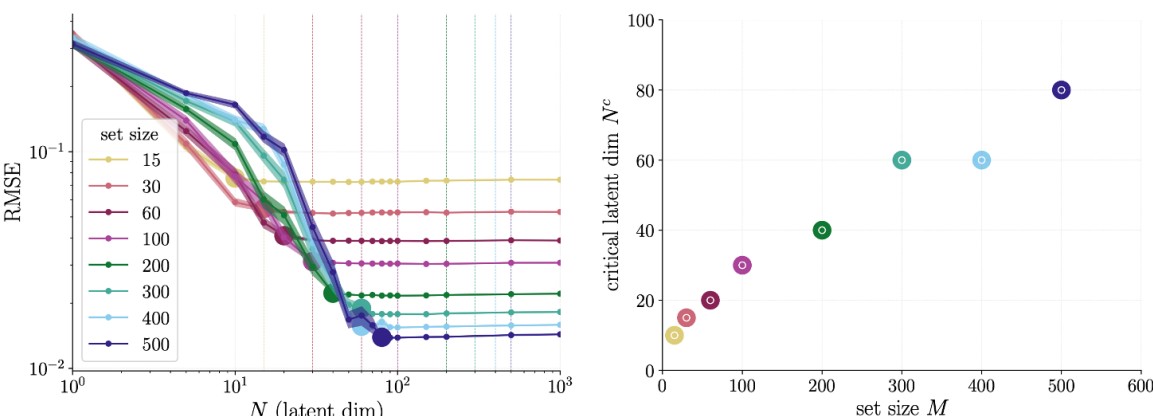

Figure 4: Illustrative toy example, from Figure 3 of Wagstaff et al. (2019). Right panel: Test performance of Deep Sets on median estimation depending on the latent dimension, and dashed lines indicate $N = M$. Here, RMSE is the Root Mean Squared Error. Left panel: Extracted critical points, and the colored data points depict minimum latent dimension for optimal performance for different set sizes. These figures are cited to show experimental results that confirm Theorem 5.7.

Furthermore, Wagstaff et al. (2022) provides a more precise analysis.

**Theorem 5.7** (Wagstaff et al. (2022)). Let $M, N \in \mathbb{N}$, with $M > N$. Then, there exists continuous permutation-invariant functions $f \colon \mathbb{R}^M \to \mathbb{R}$ which are not continuously sum-decomposable via $\mathbb{R}^N$.

This implies that for Deep Sets to be capable of representing arbitrary continuous functions on sets of size $M$, the dimension of the latent space $N$ must be at least $M$. A similar statement is also true for models based on max-decomposition, such as PointNet (Qi et al., 2017a).

**Definition 5.5.** The function $f$ is said to be max-decomposable if there are functions $\rho$ and $\phi$ such that

$$f(\mathcal{S}) = \rho \left( \max_{\boldsymbol{s} \in \mathcal{S}}(\phi(\boldsymbol{s})) \right), \tag{18}$$

where max is taken over each dimension independently in the latent space.

**Theorem 5.8.** Let $M > N \in \mathbb{N}$. Then there exist continuous permutation-invariant functions $f \colon \mathbb{R}^M \to \mathbb{R}$ which are not max-decomposable via $\mathbb{R}^N$.

Figure 4 shows the illustrative example for the above theorems. Theorem 5.7 implies that the number of input elements $M$ has an influence on the required latent dimension $N$. The neural network, which has the architecture of Deep Sets, is trained to predict the median of a set of values. The input sets are randomly drawn from either a uniform, a Gaussian, or a Gamma distribution. This figure shows the relationship between different latent dimensions $N$, the input set size $M$, and the predictive performance, and it can be seen that

- The error monotonically decreases with the latent space dimension for every set size;

- Once a specific point is surpassed (referred to as the critical point), enlarging the dimension of the latent space no longer leads to a further reduction in error;

- As the set size increases, the latent dimension at the critical point also grows.

It should be noted that the critical points are observed when $N < M$. The reason behind this phenomenon lies in the fact that the models do not acquire an algorithmic solution for computing the median. Instead, they learn to estimate it based on samples drawn from the input distribution encountered during training.

### 5.2.1 The Choice of Aggregation

The Deep Set architecture exhibits invariance due to the inherent invariance of the aggregation function). Theoretical justification for summing the embeddings $\phi(s)$ is provided by the sum-decomposability (see Section 5.1 for more details). In practice, mean or max-pooling operations are commonly employed, offering simplicity and invariance as well as numerical advantages for handling varying population sizes and controlling input magnitude for downstream layers. This section explores alternative approaches and their respective properties.

**Proposition 5.9** (Sum Isomorphism (Soelch et al., 2019)). *Theorem 5.3 can be extended to aggregations of the form $\alpha_g = g \circ \sum \circ g^{-1}$, i. e. summations in an isomorphic space.*

*Proof.* From $\rho \circ \sum \circ \phi = (\rho \circ g^{-1}) \circ g \circ \sum \circ g^{-1} \circ (g \circ \phi)$, sum decompositions can be constructed from $\alpha_g$-decompositions and vice versa. $\square$

This class includes mean (with $g((s_1, \ldots, s_{n+1})) = (s_1, \ldots, s_n)/s_{n+1}$, $g^{-1}(s) = (s^\top, 1)^\top$) and logsumexp ($L\Sigma E$) with $g = \ln$. Interestingly, $L\Sigma E$ can behave as max or linear function of summation.

We can observe that divide-and-conquer operations also yield invariant aggregations. Here, Soelch et al. (2019) claims as the commutative and associative binary operations like addition and multiplication yield invariant aggregations. This means that order invariance is equivalent to conquering being invariant to division. In the context of aggregation, order invariance is equivalent to the conquering step remaining invariant to division. This concept extends beyond the realm of basic arithmetic operations and includes logical operators such as any or all, as well as sorting operations that generalize max, min, and percentiles like the median. While these sophisticated aggregations may not be practical for typical first-order optimization, it is worth noting that aggregation techniques can encompass a wide range of complexities. Soelch et al. (2019) propose the learnable aggregation functions, namely recurrent aggregations.

**Definition 5.6** (Recurrent and Query Aggregation (Soelch et al., 2019)). *A recurrent aggregation is a function $f(\mathcal{S}) = a$ that can be written recursively as:*

$$q_t = \text{query}(q_{t-1}, a_{t-1})$$
$$\hat{w}_{i,t} = \text{attention}(m_i, q_t)$$
$$w_t = \text{normalize}(\hat{w}_t)$$
$$a_t = \text{reduce}(\{w_{i,t}, m_i\})$$
$$a = g(a_{1:T}),$$

*where $m_i = \phi(s_i)$ is an embedding of the input population $\{s_i\}$ and $q_1$ is a constant.*

If the reduced operation remains invariant and the normalized operation is equivariant, both recurrent and query aggregations maintain invariance (see Figure 5). Empirical studies show that learnable aggregation functions introduced in this work are more robust in their performance and more consistent in their estimates with growing population sizes. The key insight of this work is that Deep Sets is highly sensitive to the choice of aggregation functions. From this observation, we propose a special class of Deep Sets that has not been studied before in Section 7.

## 6 Datasets

In this section, we introduce some commonly used datasets for evaluating the performance of neural networks that approximate set functions.

**Flow-RBC (Zhang et al., 2022c):** The Flow-RBC dataset comprises 98,240 training examples and 23,104 test examples. Each input set represents the distribution of 1,000 red blood cells (RBCs). Each RBC is characterized by volume and hemoglobin content measurements. The task involves regression, aiming to predict the corresponding hematocrit level measured on the same blood sample. In a blood sample, there are various components, including red blood cells, white blood cells, platelets, and plasma. The hematocrit level quantifies the percentage of volume occupied by red blood cells in the blood sample.

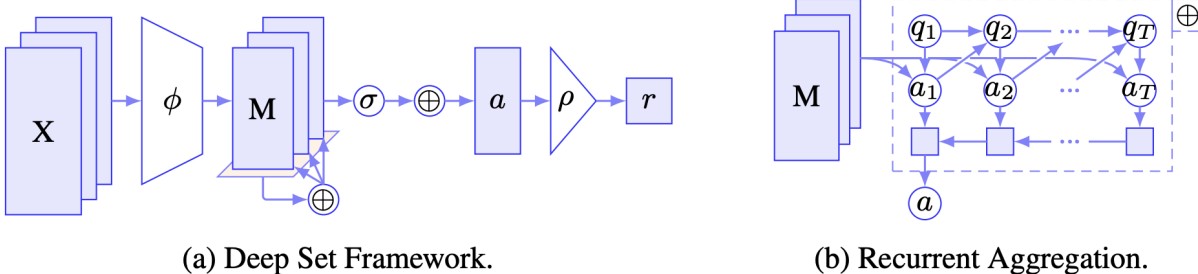

(a) Deep Set Framework.                    (b) Recurrent Aggregation.

Figure 5: Deep Set architecture and Recurrent aggregation function from Figure 1 of Soelch et al. (2019). This figure is cited to give an overview of their proposed Recurrent aggregation module.

**Celebrity Together dataset (Zhong et al., 2018):**   The Celebrity Together dataset consists of images depicting multiple celebrities together, making it a suitable choice for evaluating set retrieval methods. Unlike other face datasets that only include individual face crops, Celebrity Together comprises full images with multiple labeled faces. The dataset contains a total of 194k images and 546k faces, with an average of 2.8 faces per image.

**SHIFT15M (Kimura et al., 2023)**   SHIFT15M is a dataset designed specifically for assessing models in set-to-set matching scenarios, considering distribution shift assumptions. It allows for evaluating model performance across different levels of dataset shifts by adjusting the magnitude. The dataset contains a total of 2.5m sets and 15m fashion items.

**CLEVR (Johnson et al., 2017):**   CLEVR dataset is a synthetic Visual Question Answering dataset. It contains images of 3D-rendered objects; each image comes with a number of highly compositional questions that fall into different categories. Those categories fall into 5 classes of tasks: Exist, Count, Compare Integer, Query Attribute, and Compare Attribute. The CLEVR dataset consists of: a training set of 70k images and 700k questions, a validation set of 15k images and 150k questions, a test set of 15k images and 150k questions about objects, answers, scene graphs, and functional programs for all train and validation images and questions. Each object present in the scene, aside from position, is characterized by a set of four attributes: 2 sizes: large, and small, 3 shapes: square, cylinder, and sphere, 2 material types: rubber, and metal, 8 color types: gray, blue, brown, yellow, red, green, purple, cyan, resulting in 96 unique combinations.

**ShapeNet (Chang et al., 2015):**   ShapeNet is a large scale repository for 3D CAD models. The repository contains over 300M models with 220,000 classified into 3,135 classes arranged using WordNet hypernym-hyponym relationships. ShapeNet Parts subset contains 31,693 meshes categorized into 16 common object classes (i.e. table, chair, plane, etc.). Each shape's ground truth contains 2-5 parts (with a total of 50 part classes).

**ModelNet40 (Wu et al., 2015):**   ModelNet40 dataset contains 12,311 pre-aligned shapes from 40 categories, which are split into 9,843 for training and 2,468 for testing.

**3D MNIST (Xu et al., 2016):**   3D MNIST is a 3D version of the MNIST database of handwritten digits. This dataset contains 3D point clouds generated from the original images of the MNIST dataset, and it contains 6,000 instances.

# 7 A Special Class of Deep Sets: Hölder's Power Deep Sets

As mentioned in the previous sections, Deep Sets and PointNet can be generalized as differences in aggregation functions. Here we consider the following quasi-arithmetic mean:

$$M_f(x_1, x_2, \ldots, x_n) := f^{-1}\left(\frac{f(x_1) + f(x_2) + \cdots + f(x_n)}{n}\right), \tag{19}$$

for $n$ numbers and $f$ is the injective and continuous function. In particular, let $x_1, x_2, \ldots, x_n$ be positive numbers and $f(\cdot) = (\cdot)^p$ for $p \in \mathbb{R}$, we can write the parametric form as

$$M_p(x_1, x_2, \ldots, x_n) := \left(\frac{1}{n}\sum_{i=1}^{n} x_i^p\right)^{1/p}. \tag{20}$$

The function $M^p$ is called the power mean or Hölder mean (Bickel & Doksum, 2015), and has many applications in machine learning (Van Erven & Harremos, 2014; Kimura & Hino, 2021; Kimura, 2021; Kimura & Hino, 2022; Oh & Kwak, 2016; Nelson, 2017). By taking specific values of $p$, we have the following special cases:

- $M_{-\infty}(x_1, x_2, \ldots, x_n) = \lim_{p \to -\infty} M_p(x_1, x_2, \ldots, x_n) = \min\{x_1, x_2, \ldots, x_n\}$.

- $M_{-1}(x_1, x_2, \ldots, x_n) = n/(\frac{1}{x_1} + \frac{1}{x_2} + \cdots + \frac{1}{x_n})$ (harmonic mean).

- $M_0(x_1, x_2, \ldots, x_n) = \lim_{p \to \infty}(x_1, x_2, \ldots, x_n) = \sqrt[n]{x_1 + x_2 + \cdots + x_n}$ (geometric mean).

- $M_1(x_1, x_2, \ldots, x_n) = (x_1 + x_2 + \cdots + x_n)/n$ (arithmetic mean).

- $M_{+\infty} = \lim_{p \to +\infty}(x_1, x_2, \ldots, x_n) = \max\{x_1, x_2, \ldots, x_n\}$.

By using this function, we can define the generalized Deep Sets as follows.

**Definition 7.1** (Hölder's Power Deep Sets). For a set $\mathcal{S} \subseteq \mathcal{V}$ and $\phi \colon \mathbb{R}^d \to \mathbb{R}_+$, we can define the Hölder's Power Deep Sets is defined as

$$f_{\mathrm{HP-DS}}(\mathcal{S}) := \rho\left(M_p^\phi(\mathcal{S})\right), \tag{21}$$

where

$$M_p^\phi(\mathcal{S}) = \left(\frac{1}{|\mathcal{S}|}\sum_{\boldsymbol{s} \in \mathcal{S}} \phi(\boldsymbol{s})^p\right)^{1/p}. \tag{22}$$

We can see that the Normalized Deep Sets and the PointNet are the special cases with $p = 1$ and $p = +\infty$, respectively.

## 7.1 Advantages of parametric form

In our Hölder's Power Deep Sets, choosing a power exponent $p$ is equivalent to choosing an architecture such as Deep Sets or PointNet. The most straightforward way to obtain the best estimation is to compare the performance of several of these architectures. Then, for this formulation, we make the following expectations.

- i) The optimal parameter $p$ depends on the dataset and problem setting.

- ii) With a good choice of $p$, better performance can be achieved than with Deep Sets or PointNet.

In the following numerical experiments, we confirm these expectations.

Table 2: Experimental results for different values of $p$ in the Hölder's Power Deep Sets. Here, $p = 1$ and $p = +\infty$ give Deep Sets and PointNet, respectively.

| dataset | $p = -1$ | $p = 0$ | $p = 1$ (Deep Sets) | $p = 2$ | $p = 3$ | $p = +\infty$ (PointNet) |
|---|---|---|---|---|---|---|
| CelebA ($\uparrow$) | 0.871($\pm$**0.003**) | 0.863($\pm$0.009) | **0.873**($\pm$0.008) | 0.863($\pm$0.004) | 0.867($\pm$0.012) | 0.862($\pm$0.004) |
| 3D MNIST ($\uparrow$) | 0.746($\pm$0.011) | 0.809($\pm$0.003) | 0.810($\pm$0.005) | **0.824**($\pm$0.003) | 0.820($\pm$**0.002**) | 0.823($\pm$0.015) |
| Flow-RBC ($\downarrow$) | 32.18($\pm$2.855) | 30.59($\pm$0.7857) | 29.58($\pm$0.997) | 28.36($\pm$**0.179**) | **28.34**($\pm$0.301) | 45.84($\pm$15.34) |

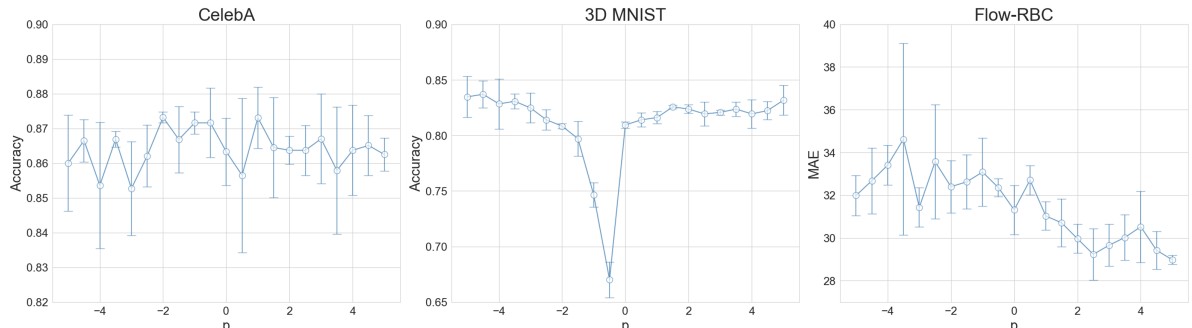

Figure 6: Performance evaluations with $p \in [-5, 5]$.

## 7.2 Numerical experiments on optimization of generalized Deep Sets

Next, we introduce the experimental results for our generalized Deep Sets.

### 7.2.1 Experimental settings

**Datasets**   In our numerical experiments, we use three datasets, CelebA (Zhong et al., 2018), 3D MNIST (Xu et al., 2016) and Flow-RBC (Zhang et al., 2022c). See Section 6 for the overview of each dataset. The performance evaluation on CelebA and 3D MNIST is accuracy, while the performance evaluation on Flow-RBC is the mean absolute error. All experiments are conducted on three trials with different random seeds, and the means and standard deviations of the evaluations are reported.

**Architecture of neural networks**   In our experiments, we use simple Deep Sets (Zaheer et al., 2017) consisting of three layers. For this base architecture, we apply the generalized aggregation function in Eq. equation 20.

**Computing resources**   Our experiments use a 16-core CPU with 60 GB memory and a single NVIDIA Tesla T4 GPU.

### 7.2.2 Experimental results

First, we confirm that changes in the power exponent $p$ in our formulation have an effect on performance. Table 2 shows the experimental results for different values of $p$ in the Hölder's Power Deep Sets. We use $p = -1, 0, 1, 2, 3, +\infty$ in this experiments, and they correspond to the harmonic mean, geometric mean, arithmetic mean, quadratic mean, cubic mean and maximum. Here, $p = 1$ and $p = +\infty$ give Deep Sets and PointNet, respectively. This experiment shows that different $p$ achieve different evaluation performances. In particular, we can observe that the parameters that achieve the best performance in terms of bias differ from those that are best in terms of variance.   More precisely, Figure 6 shows the performance evaluations with $p \in [-5, 5]$. Experimental results are reported for different values of $p$ for every 0.5 within this range. From this figure, we can see that the optimal parameter $p$ depends on the dataset, and it does not necessarily correspond to well-known special cases such as Deep Sets or PointNet.

Table 3: Optimization of power exponent for the Hölder's Power Deep Sets. Each cell shows the obtained $p$, the achieved evaluation performance and the required computation time, respectively.

| dataset | linear search | Bayesian optimization | gradient descent |
|---|---|---|---|
| CelebA ($\uparrow$) | $p = -2.000$ 
 $0.873(\pm0.002)$ 
 $120,000$ [sec] | $p = -0.521$ 
 $0.867(\pm0.015)$ 
 $33,000$ [sec] | $p = 0.106$ 
 $0.870(\pm0.009)$ 
 $3,000$ [sec] |
| 3D MNIST ($\uparrow$) | $p = -5.500$ 
 $0.843(\pm0.006)$ 
 $32,000$ [sec] | $p = -6.405$ 
 $0.842(\pm0.004)$ 
 $3,200$ [sec] | $p = 0.091$ 
 $0.822(\pm0.011)$ 
 $800$ [sec] |
| Flow-RBC ($\downarrow$) | $p = 3.000$ 
 $28.34(\pm0.301)$ 
 $280,800$ [sec] | $p = -0.374$ 
 $33.54(\pm0.441)$ 
 $203,580$ [sec] | $p = 2.850$ 
 $30.32(\pm2.685)$ 
 $7,020$ [sec] |

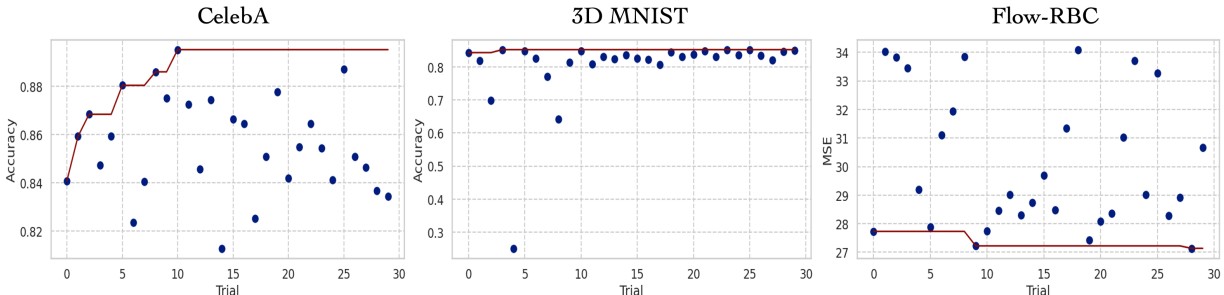

Figure 7: Bayesian optimization history for each dataset.

The above experimental results confirm expectations i) and ii). The next question is how to find a good parameter $p$ efficiently. Enumerating several special forms and comparing their performance, as in the experiment in Table 1, is the simplest way to do this, but it seems to provide too few choices. An alternative way is to try all the values of $p$ within a certain range, as shown in Figure 6, but this is computationally expensive when the search range is wide or when small changes in $p$ are examined. To solve these problems, we consider directly optimizing the power exponent $p$ using Bayesian optimization and gradient descent. In this experiments, we set $p \in [-10, 10]$. For linear search, we compare the performance with different $p$ for every 0.5 within this range. For Bayesian optimization, we use the Optuna (Akiba et al., 2019) and set the number of trials to 30. For the gradient descent, we use the Adam optimizer (Kingma & Ba, 2014) with $lr = 0.001$.

Table 3 shows the experimental results on the optimization of power exponent $p$ for the Hölder's Power Deep Sets. In this table, we report the obtained $p$, the achieved evaluation performance and the required computation time. In addition, Fig. 7 shows the history of Bayesian optimization for each dataset. From these results, we can see that comparable performance to linear search can be achieved in reasonable computation time by Bayesian optimization and gradient method.

### 7.3   Limitations of Hölder's Power Deep Sets

Here we discuss several limitations of Hölder's Power Deep Sets. Our architecture leverages the concept of the quasi-arithmetic mean to connect Deep Sets and PointNet in the linear case. However, in order to establish the practical advantages, thorough experimental and theoretical analysis is required when nonlinear $\rho$ is used.

# 8 Conclusion and discussion

Unlike typical machine learning models that handle vector data, when dealing with set data, it is crucial to ensure permutation-invariance. In this survey, we discussed various neural network architectures that satisfy permutation-invariance and how they are beneficial for a range of tasks. Permutation-invariant architectures not only enhance the ability of models to learn from and make predictions on set data but also open the door to more sophisticated handling of complex structures in machine learning. As we delve deeper into the potential of permutation-invariant neural networks and explore their adaptations for specific tasks, future research will likely focus on refining these models, addressing challenges, and uncovering novel applications. This dynamic landscape promises further advancements in machine learning, bridging the gap between vector and set data to unlock new opportunities for understanding and processing complex information structures.

## 8.1 Future directions

Future research topics for machine learning models that deal with sets or permutation-invariant vectors may include the following.

- **Research in specific areas is still insufficient**: For example, there is limited work on XAI and federated learning for neural networks dealing with sets mentioned in Section 4.8. The connection between subfields, which are well studied in general machine learning, and models that approximate set functions is meaningful.

- **Availability of more datasets**: As reviewed in Section 6, there are many available set datasets for machine learning, but they are still underdeveloped compared to computer vision and natural language processing. In particular, a de-facto standard dataset of the scale of ImageNet (Deng et al., 2009; Krizhevsky et al., 2012; Russakovsky et al., 2015) in image classification would contribute greatly to the development of the field.

We also consider the following research questions concerning our generalized Deep Sets as follows.

- **Efficient parameter optimization**: The range of parameters introduced in our generalized Deep Sets is $\mathbb{R}$. In our experiments, however, we have a finite number of candidates or ranges and optimize within them. We expect that the usefulness of our formulation would be enhanced if a wider range of parameters could be searched efficiently.

- **Further generalization**: Our Hölder's Power Deep Sets given in Eq. equation 20 can be further generalized in the form of weighted mean as $(\frac{1}{|S|} \sum_{s \in \mathcal{S}} w(s) \cdot \phi(s)^p)^{1/p}$, with $\sum_{s \in \mathcal{S}} w(s) = 1$. Considering such a generalization allows for more freedom in the model and may allow for further improvements.

- **Theoretical analysis**: In this work, we investigated the behavior of generalized Deep Sets from numerical experiments. The theoretical analysis of this formulation is expected to contribute to further research.

- **Comprehensive experiment**: We performed numerical experiments on three different datasets. Experimental results show that the optimal parameters vary depending on the dataset. We could potentially obtain further insights by conducting similar experiments on a larger number of datasets.

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
