# OpenReview forum: "On permutation-invariant neural networks: A survey"
_TMLR — Rejected by TMLR_

### Review · Reviewer_wi7p · 2023-12-08

**Summary Of Contributions:**

The manuscript provides a survey of neural network-based approaches to the approximation of functions defined on sets. In particular, it highlights the importance of permutation invariance and provides an extensive list of different architectures, task set functions are used for as well and data sets commonly used for the evaluation of proposed models. Further, the manuscript gives an overview of central results in the representation and approximation of set functions via deep sets and point nets, which correspond to sum and max decomposable functions, respectively.

**Audience:**

Yes

**Broader Impact Concerns:**

None.

**Claims And Evidence:**

No

**Requested Changes:**

**Changes required before consideration for publication.** The following aspects have to be adjusted to be considered for publication:

1. **Novelty.** As elaborated above, the manuscript lacks novelty in my opinion. Here is a collection of possible extensions, where we do not expect improvements in all, but at least one of the listed aspects, or the addition of another original contribution:

    * *Extension of existing approximation results.* One way to contribute to the field would be to provide new approximation theoretical results. One particular shortcoming of existing results that we currently see is that do not come with any structural assumptions on $\phi$. In particular, it would be interesting to investigate the representation and approximation properties of deep sets and point nets when $\phi$ is computed by a fixed neural network architecture. Another theoretical contribution could be to provide a separation result between deep sets and point nets.
    * *New empirical results.* Another insight could be provided by empirical evidence on the efficacy of different architectures or the suggestions of novel architectures.
    * *Novel perspective.* A novel perspective or a theory unifying existing results could provide an original contribution.
    * *List of open problems:* A list of open problems can help accelerating and guiding research on a topic. A particularly well-suited place for such a collection is a survey on the topic. Hence, if you could offer a more comprehensive presentation regarding existing results, with that I mean including insight to the proofs, and wrap things up with a list of important directions and open problems, the survey could serve as a guide for people new to the topic.

2. **Usefulness as a survey.**
    * *Focus and completeness.* If the focus of the survey is on the approximation aspects of neural set functions, then this should be reflected in the organization of the manuscript as well as the length of the respective sections. Currently, there is only one of given sections (not counting intro and discussion) devoted to the approximation properties. Further, if the approximation capabilities are the focus of the survey, they should be more comprehensive giving insight into the main proof ideas and stating the approximation results concerning the Hausdorff distance.

    * *Insights to the proofs.* One benefit of a good survey, in particular for people entering a field, is that one spares reading a variety of papers to understand the results as well as the used proof techniques. However, the proof of Theorem 5.2 should be possible just by examining the definition of Janossy representations. Overall, the survey would be much more helpful if it would give

    * *Strength of statemets.* Finally, some of the theoretical results are stated in an unnecessarily weak form. In particular, Theorem 5.3 in its original form due to Zaheer et al. states that a function is sum-decomposable if and only if it is a set function. Further, Proposition 4.1 can be extended to an equivalence as well.

3. **Mathematical accuracy.**
At various points, the manuscript leaves room for mathematical ambiguity, which considerably slows down reading and can lead to *compilation errors*. In particular, we want to mention the following aspects:

     * *Definition of permutation invariance.* As mentioned above, I do not understand the definition of permutation invariance given in the manuscript. By definition, for a function $f\colon 2^\mathcal V\to\mathbb R$ the value of the function does not depend on the ordering of the set, as in the notation of the manuscript, $\pi_\mathcal SS = S$ in a set-theoretic sense. I do understand that when working with tuples, i.e., ordered sets, it makes sense to talk about permutation invariance and that permutation invariance is required there to perceive the function as a set function rather than as a function defined on the tuple. On the other hand, in Definition 5.5 of Janossy pooling, a *permutation-sensitive* function $\Phi$ is considered, where it is not quite clear what this means (it is also not defined) as it can not be a function of the set then. One way would be to let $\Phi$ be defined on tuples, so either $\Phi\colon\cup_{k=1}^{\lvert \mathcal V \rvert} \mathcal V^k\to\mathbb R$ or $\Phi\colon \mathcal V^{\lvert \mathcal V\rvert} \to \mathbb R$. To prevent confusion and as the manuscript puts a focus on permutation invariance, it is important to be mathematically precise here. Additionally, it would make sense to provide an example of a permutation-sensitive function as all considered architectures are naturally defined on the set and are therefore permutation invariant.
    There are various possibilities to clear up this confusion, one of them being the following: One could refer to functions $f\colon 2^\mathcal V\to\mathbb R$ as *set functions* and consider functions defined on tuples $\Phi\colon\cup_{k=1}^{\lvert \mathcal V \rvert} \mathcal V^k\to\mathbb R$. Then for $\Phi$ one can define what it means to be permutation invariant and then the statement is that $\Phi$ can be perceived as a set function if and only if $\Phi$ is permutation invariant. Conversely, a set function $f$ induces a pertmutation invariant function $\Phi$ on tuples.
    This, however, is only one way to go and is eventually up to the authors.
     * *Missing definitions.* The notions of permutation-sensitive and $k$-ary Janossy representations are not defined.
     * *Collection of ambiguities.* In various statements the mathematical writing is not entirely clear. Here is a (not necessarily exhaustive) list of them:
        * Definition 2.1 is not a definition, rather a convention of notation.
        * In Definition 2.2 it is not clear what $[1, \lvert \mathcal V\rvert]$ denotes. This is the standard notation for the interval from $1$ to $\lvert\mathcal V\rvert$, but rather seems to denote the set $\mathcal V$ or $\{1, \dots, \lvert\mathcal V\rvert\}$. If however, $\mathcal V = \{1, \dots, \lvert\mathcal V\rvert\}$, then $s_i\in\mathcal V$ is an integer rather than an element in $\mathbb R^d$ and hence the definition $\varphi(s_i)=s_i$ is not clear. In Definition 2.2 any set is identified with its representation in $\mathcal V\subseteq\mathbb R^d$, which causes confusion as by Definition 2.1 $\mathcal V\subseteq\mathbb R$. In general, I think it is necessary to clear up the notation and conventions here.
        * *Proposition 4.2.* It is not clear from the statement that $\rho(x)=x$ and $\phi\ge0$ is an assumption. Further, the correct way of stating i) and ii) would rather be *The function $f\colon 2^\mathcal V\to\mathbb R$ computed by a deep set is a modular function.* rather than *the modular function*.
        * *Theorem 5.6.* It is not clear what differentiability for a set function means and where the definition of the gradient norm comes from. Please define the notion of derivatives of a set function.
       * *Notation of tuples.* The notation of tuples used in Equation (3) is not defined.

**Changes suggested before consideration for publication:** We strongly recommend to adjust the following points before consideration for publication:

* *Proof of Proposition 4.2.* Here, we sketch a very simple proof as in its current form we found it to not be very easy to follow. In particular, the equivalence of the checked property to the submodularity is not clear and no reference for it is given. Let us set $a\coloneqq \max_{s\in\mathcal S} \phi(s), b\coloneqq \max_{s\in\mathcal T} \phi(s), c\coloneqq \max_{s\in\mathcal S\cap \mathcal T} \phi(s)$ and $d\coloneqq \max_{s\in\mathcal S\cup\mathcal T} \phi(s)$, then we need to show $a+b\ge c+d$. Note however, that $a, b \ge c$ and thus $\min(a,b) \ge c$ as well as $d=\max(a,b)$. Thus, we estimate $a+b = \min(a,b) + \max(a,b) \ge c+d$.

* *Introduction.* In the introduction the outline of the manuscript is presented twice: One time as a running text, one time as a list. I would recommend only doing one.

* *Indicate contributions.* One should not overstate the contributions of a manuscript, but it makes sense to explain what readers get from the manuscript.

* *Permutation equivariance.* The manuscript mentions equivariance multiple times, so I would recommend providing a definition to have everyone on board.

* *Ambiguous writing and unclear claims.* There are multiple instances, where writing is unclear, here is a list of things, I found ambiguous or unclear:
    * Before the beginning of Section 3 it is stated that *By satisfying these properties [permuation invariance],
neural networks can effectively model and approximate set functions*. This is not clear to me. Only because a function is permutation invariant it is not clear why it should be able to approximate arbitrary set functions.
    * In Subsection 3.4 it is not clear what *superior performance* means, in particular it is not clear what the compared models are.
    * In Subsection 3.4 it is stated that *Furthermore, this paper also releases a dataset ...* where it is not clear what *that paper* refers to.
    * In the paragraph regarding minimum assignment it is not defined what LSP is and it is not clear what it means to *ensure convergence*.
    * After Definition 5.4 it is stated that *Achieving permutation invariance poses a fundamental challenge in the design of models, as it requires finding the right trade-off between expressive power and maintaining the desired property*. In the light that any set function is permutation invariant, I do not agree with the existence of this trade-off. Rather if we want to design set functions, we need to design them in a way that they actually only depend on the set.
    * In Definition 5.5 it is stated that *and this is the form of sum-decomposable 5.1, and permutation-invariant 2.5*. This is rather sloppy and I would recommend to remark after the definition that both sum-decomposable and permutation-invariant functions can be seen as special cases of Janossy pooling. This, however, is only obvious to me if we consider $k$-ary Janossy pooling that you did not introduce.
    *  After Theorem 5.2 in iii) it is stated that $\mathcal S_{\{k\}}$ denotes the set of all $k$-tuples. From my understanding this should denote the set of all $k$-subsets, which is a smaller set as there are $\lvert\mathcal V\rvert^k$ tuples.
    * In the proof of Theorem 5.3 it is not clear what $\Phi$ is
    * All figures should be referenced in the text.
    * Figure 5 is not explained in sufficient detail: It is also left and right rather than top and bottom panel; further, it is not clear what RMSE is here, whether this is for deep sets or point nets and why the model achieves a lower RMSE for larger sets. Finally, it should be discussed after Theorem 5.7 rather than after Theorem 5.8.
    * It is not clear what the benefit of the approach in Subsection 5.3.1 is.
    * It is not defined what *divide-and-conquer operations* are.
    * In the conclusion it is stated that the manuscript *introduced various neural network architectures*. This sounds like the manuscript proposed these architectures, so I would recommend changing this to *discusses / presents / ... various neural network architectures*.


* *Typos and minor writing things.* There is a variety of typos that should be corrected. In particular regarding the third person singular -s. Here is a (not necessarily exhaustive list):
   * At the beginning of Section 3 I would recommend to change *we overview the neural network architectures* to *we give an overview of neural network architectures* since there might be more architectures.
    * At the beginning of Section 3 the ending of the second sentence is not perfect sense, maybe it can be changed to *as well as the corresponding applications.*
    * At the beginning of Section 3, I recommend adjusting to *In particular, we focus on architectures following **the idea of** Deep Sets,*.
    * In Subsection 3.1 it is stated that *it is obvious that Deep Sets [...] can approximate the set function by using arbitrary neural networks $\phi$ and $\rho$.* as well as *It is
also known that Deep Sets has the universality*. Firstly, this is not obvious to me and at that point it has been explicitly defined what it means to approximate a set function not what universality means in this context.
    * In Subsection 3.2 I would recommend changing the sentence to ***A** function that can be written [..] is referred to as a sum-decomposable...*.
    * In Subsection 3.3 it should be *Set Transformer (Lee et al., 2019) utilize the* as well as *Set Transformer address*.
    * In Proposition 4.1 $\psi$ should be $\phi\colon\mathcal V\to\mathbb R$.
    * *Sinkhorn* should be in capital letters as a name.
    * In Subsection 4.8 it should be *Cuturi (2013) propose**s***.
    * In Definition 5.4 it should be *sum-decomposition* and I would recommend to change it to *$(\phi, \rho)$ is said to be a ...*.
    * Both $f_{\text{Normalized-DeepSets}}$ and $f_{\text{N-DeepSets}}$ is used.
    * In the conclusion the sentence ending *... learning, such as graph.* has to be corrected.

* *Formatting.* Just some nitpicky comments, feel free to ignore them.

    * It is recommended to use \colon instead of : when writing functions, i.e., $f\colon X\to Y$ vs $f: X\to Y$.
    * I would not recommend using both \varphi and \phi as they are the same letter, but that is a style choice.
    * In Definition 4.1, Definition 4.2 etc. I would recommend to write $f\colon 2^\mathcal V\to\mathbb R$ as this makes the setting unambiguous.
    * It could be nice to shorten notation $f_{\text{DeepSets}}$ to $f_{\text{DS}}$ and so on.

**Strengths And Weaknesses:**

The manuscript focuses on the timely and important area of set functions and provides an extensive overview over the current literature.
Whereas I value good expository articles, the following aspects constitute weaknesses of the manuscript, where I will make specific suggestions for changes in the according field:

1. **Novelty.** My most severe concern is that I don’t see any new results in the manuscript. As such, there are neither new theoretical nor empirical results and also no novel perspective on existing results.

2. **Usefulness as a survey.** Judging from the title of the manuscript, its focus lies on the approximation aspects of neural network-based set functions and in particular the role of permutation invariance. Where I see a great potential benefit of such a survey the following aspects limit the usefulness of the survey in its current form:

     * *Listing architectures and tasks.* A lot of space is devoted to a comprehensive discussion of different architectures, tasks, or datasets. However, these architectures and tasks are described in such little detail that one can merely get a rough idea of them rather than an explicit understanding of them. While this is not necessarily bad, it does take up a considerable portion of the manuscript while not contributing to the understanding of the approximation properties, which is the main focus of the manuscript.
     * *Insight gained from the approximation section.* A similar thing applies to the section regarding the approximation results: It consists of a mere listing of results from 4 papers without putting them into a novel relation to another, extending them, or giving valuable insights into the proofs. Given that all these papers are well written and 4 is a relatively moderate number of papers, I wonder what the benefit of this section is. Further, I feel that the current presentation would be stronger if it would provide precise statements mentioned in Subsection 5.2.

3. **Mathematical accuracy.**  For a survey article to be useful for people entering the field, it is important to be very precise and not assume any terminology. Unfortunately, there are various aspects, where the mathematical writing is inaccurate, a complete list of these can be found in the following field. However, there is one aspect that I want to highlight here, which is the definition of the permutation invariance: Any set function, i.e., any function defined on the powerset of a set, does by definition not depend on the ordering within the set, see also **Property 1**  in Zaheer et. al. (2017). Hence, I suspect that the problem of permutation invariance might come into play when realizing a set function by a function acting on an ordered list of elements, i.e., a tuple. In its current form, however, the definition of permutation invariance does not make sense from a purely mathematical standpoint.

4. **Organization of the paper and quality of writing.** Various cosmetic aspects of the paper are not satisfactory:
     * Title is somewhat misleading in its current form: approximation does not seem to be the focus, it is rather a general survey / list of papers in the area; also the title suggests that the permutation-invariant perspective is unique to this survey, but it is present in all approximation theoretic works in this area
     * A general discussion of different types of approximation and representation at the beginning of the paper or Section 5 would be helpful. In particular, I see at least three types of results: representation, uniform approximation (e.g., Theorem 5.6) and approximation for Hausdorff (Wasserstein) metrics.
     * Subsection headers are sometimes misleading as 5.2 is called **Expressive power of Deep Sets and PointNet** and 5.3 is called **Sufficient and necessary conditions for Deep Sets to be universal**. However, in Subsection 5.3 results regarding both the representational capacity as well as the approximation power of deep sets and point nets are presented.
     * Some theorems are neither proven nor cited.
     * Subsection 4.6 feels like is oddly located in Section 4, which is otherwise just a list of tasks. It would make much more sense to move it to Section 5 as it shows the equivalence of modular functions and sum decomposable functions with $\rho(x)=x$.
     * Proof of Prop 4.2 is unreasonably complicated, I provided a much shorter version in the next field.
     * Section 6 is oddly placed, it would make much more sense to discuss it in
     * Often, terminology is used that was not defined previously. For a list of such cases, see the next field.
     * Overall, the read is not smooth, which comes through typos and a mere listing of definitions and theorems without putting them into perspective.

Overall, I feel that in its current form, the manuscript should not be considered for publication. The main reason for this is its lack of novelty and its limited usefulness as a survey. In particular, regarding the usefulness we think that for people familiar with the topic, it does not add any benefit or new perspective and for people new to the field it does not offer enough substance to equip people with the tools required to start research in this area.

In the following field, we make precise suggestions on how to improve on these aspects.

---

> ### Author Response · Authors · 2024-02-04
> **Response from the authors.**
>
> We appreciate your very constructive review.
>
> We believe that a revised manuscript based on your suggestions will contribute to the development of this research field.
>
> Novelty
>
> As you mentioned, additional contributions based on the survey are very important.
>
> Based on your examples, we have considered and proposed a new generalization of Deep Sets in Section 7.
>
> From the survey, we can see that Deep Sets and PointNet can be generalized to different aggregation functions, and their behavior is highly dependent on the choice of aggregation function.
>
> We therefore realized that Deep Sets and PointNet can be generalized using power mean, a parametric special case of the quasi-arithmetic mean.
>
> Using such a parametric generalization, we expected that a good optimization of this generalization parameter would lead to improved performance.
>
> In addition, we listed open problems in Section 8.
>
> Usefulness as a survey
>
> > Focus and completeness
>
> As you pointed out, there was a discrepancy between the title and the content of our manuscript.
>
> To address this, we modified the title of the manuscript to align with the content.
>
> > Insights to the proofs.
>
> We added that the proof of Theorem 2 is a confirmation of the definition of Janossy pooling, as you point out.
>
> > Strength of statemets.
>
> We have rewritten it to fit Theorem 5.3's original claims.
>
> Mathematical accuracy
>
> > Definition of permutation invariance
>
> We have modified the definition of permutation-invariance in the form you recommended.
>
> > Missing definitions
>
> We added missing definitions
>
> > Collection of ambiguities
>
> We have added missing notations and revised the manuscript to resolve the ambiguity.
>
> > Proof of Proposition 4.2
>
> We modified the proof of Proposition 4.2.
>
> > Introduction
>
> We removed the redundant sentence.
>
> > Indicate contributions
>
> We listed contributions in Section 1
>
> > Permutation equivalence
>
> We have added a new section 2.2, which describes the equivariant property.
>
> > Ambiguous writing and unclear claims
>
> We have revised the manuscript throughout to resolve the ambiguity.
>
> > Typos and minor writing things
>
> We fixed typos you mentioned.
>
> > Formatting
>
> We have modified it as much as possible to be in line with the formatting as you recommend.

---

> > ### Comment · Reviewer_wi7p · 2024-03-05
> >
> > Dear authors,
> >
> > thanks for your efforts in improving the manuscript. Even in the updated form, there are still things that need to be improved before I can recommend the manuscript for publication:
> >
> > **Mathematical rigor:**
> > My most severe concern is the inaccurate use of mathematical language and concepts, which has the potential to confuse potential readers. Most importantly this includes the following aspects:
> > * *Definition of invariance:* In Definition 2.2 I have the problem that permutations only make sense for ordered sets or tuples. In Definition 2.5, it is not clear what $\Phi(\mathcal S)$ means on a formal level as $\mathcal S\subseteq\mathcal V$, but $\Phi$ is defined on tuples; however, there is no unique way how to identify subsets with tuples. In Definition 2.6 the function $\Phi$ is defined on the power set and hence by definition permutation invariant. In Definition 2.7 there are multiple problems: Firstly, equivariance only makes sense, if $f$ maps to tuples rather than to $\mathcal V$. However, if $f$ is actually defined on the power set $2^{\mathcal S}$, then it has to be permutation invariant and therefore it can not be permutation equivariant. Hence, here one would have to work with tuples. Finally, there is a typo in the index as it should be s_{\pi_{\mathcal S}(1)} instead of s_{\pi_{\mathcal S(1)}}.
> > * *Proposition 4.2.:* Still, the meaning of "Deep Sets are the modular function" and "PointNet is the submodular function" is unclear and it is unclear what $x\in S$ refers to. Further, the assumption that $\rho(x) = x$ is still missing in the statement.
> >
> > **Contributions:**
> > I have suggested the following things to provide novel contributions: Extension of existing approximation results, New experimental results, novel perspective and a list of open problems, which is partly addressed by newly added sections regarding Hölder sets and the outlook. However, I still have the following concerns. Firstly, the outlook is rather general with vague future directions rather than consisting of precise open questions. More importantly, however, the experimental section has the following shortcomings: It is claimed that Hölder sets provide a "Novel generalization of Deep Sets". However, as far as I can see it, they provide a special class of deep sets that has not been studied before; this is fine, but should be stated correctly. More importantly, it remains unclear, what the motivation of this architecture is and whether it provides any improvement over deep sets with nonlinear $\rho$. Either a theoretical motivation or a thorough empirical comparison to existing models should be given, otherwise, it remains an arbitrary proposition of a new architecture.
> >
> > Smaller things that caught my attention and that should be addressed include:
> > * *Title:*  I would suggest to remove "the" from the title
> > * Proposition 4.1. should have a citation.
> > * In Proposition 4.2. the proof for i) should be added.
> > * Definition 5.1 would make sense to be presented when introducing deep sets.
> > * Definition 5.2 could well be part of Definition 5.1
> > * Notation regarding $s_i$ and $\boldsymbol{s}_i$: I am find the identification made here unnatural and don't see why it is necessary.
> > * In Subsection 7.1: It is a bit of a stretch to call i) and ii) a conjecture.
> > * In Subsubsection 7.2.2 you state: "Here, Deep Sets and PointNet correspond to special cases with p = 1 and p = +∞, respectively." I disagree with this. It is correct that $p=1$ and $p=+\infty$ will give a Deep Set and PointNet respectively, however, the contrary is not true: not every Deep Set or PointNet can be generated this way, only the ones with $\rho(x)=x$ as far as I saw it.
> >
> > I hope these comments help you in further improving the manuscript.

---

> > > ### Author Response · Authors · 2024-03-11
> > > **Response from the authors.**
> > >
> > > We really appreciate your very helpful suggestions for improving our manuscript.
> > > We will address your suggestions step by step.

---

> > > > ### Author Response · Authors · 2024-03-11
> > > > **Response from the authors.**
> > > >
> > > > We have removed "the" from the title of the manuscript.

---

> > > > > ### Author Response · Authors · 2024-03-11
> > > > > **Response from the authors.**
> > > > >
> > > > > We have added citation to Proposition 4.1.

---

> > > > > > ### Author Response · Authors · 2024-03-11
> > > > > > **Response from the authors.**
> > > > > >
> > > > > > - We have added proof for i) to Proposition 4.2.
> > > > > > - We have clarified Proposition 4.2 statements. In addition, we have added additional assumption.

---

> > > > > > > ### Author Response · Authors · 2024-03-11
> > > > > > > **Response from the authors.**
> > > > > > >
> > > > > > > We have defined sum-decomposability in the part introducing Deep Sets.

---

> > > > > > > > ### Author Response · Authors · 2024-03-11
> > > > > > > > **Response from the authors.**
> > > > > > > >
> > > > > > > > Definition of invariance:
> > > > > > > > - In Definition 2.2, we defined permutations on tuples.
> > > > > > > > - In Definition 2.5, we defined the function on tuples.
> > > > > > > > - We have removed Definition 2.6.
> > > > > > > > - In Definition 2.7 (currently 2.6), we defined permutation equivariance on tuples. We fixed typos.

---

> > > > > > > > > ### Author Response · Authors · 2024-03-11
> > > > > > > > > **Response from the authors.**
> > > > > > > > >
> > > > > > > > > We restate H\"{o}lder's Power Deep Sets as a previously unstudied special class of Deep Sets, rather than a novel generalization.

---

> > > > > > > > > > ### Author Response · Authors · 2024-03-11
> > > > > > > > > > **Response from the authors.**
> > > > > > > > > >
> > > > > > > > > > We have changed the statement: "Deep Sets and PointNet correspond to special cases with p = 1 and p = +∞, respectively." -> "p=1 and p=+\infty give Deep Sets and PointNet, respectively."

---

> > > > > > > > > > > ### Author Response · Authors · 2024-03-11
> > > > > > > > > > > **Response from the authors.**
> > > > > > > > > > >
> > > > > > > > > > > In Subsection 7.1, we reworded 1 and 2 as expectations instead of conjectures.

---

> > > > > > > > > > > > ### Author Response · Authors · 2024-03-11
> > > > > > > > > > > > **Response from the authors.**
> > > > > > > > > > > >
> > > > > > > > > > > > We have added a subsection on Limitations since, as you mentioned, there was a lack of discussion of the architecture of our Power Deep Sets in the non-linear case.

---

> > > > > ### Comment · Reviewer_Mt8i · 2024-03-11
> > > > >
> > > > > I am sorry for interrupting discussion.  However, I do not agree with the current title.  Since this paper is a survey paper, I think the authors should explicitly show that it is a survey paper.

---

> > > > > > ### Author Response · Authors · 2024-03-11
> > > > > > **Response from the authors.**
> > > > > >
> > > > > > Many thanks for your useful comments.
> > > > > > How about the following title?
> > > > > >
> > > > > > "On permutation-invariant neural networks: a review of literature"
> > > > > >
> > > > > > If there are any additional information that should be included in the title, we would be glad if you could point it out to us.

---

> > > > > > ### Author Response · Authors · 2024-03-11
> > > > > > **Response from the authors.**
> > > > > >
> > > > > > We have changed the title as "On permutation-invariant neural networks: A survey".

---

### Review · Reviewer_Mt8i · 2024-01-05

**Summary Of Contributions:**

This work discusses several neural network-based set functions and their applications.  In addition to these, the authors provide some theoretical perspectives of such neural network-based set functions and datasets for set-related tasks.

**Audience:**

Yes

**Broader Impact Concerns:**

I do not have any concerns on broader impacts.

**Claims And Evidence:**

No

**Requested Changes:**

Please see the text box above.

**Strengths And Weaknesses:**

Strengths

* Research on set-taking neural networks is an interesting topic in machine learning.

Weaknesses

* This work fails to provide insights for neural network-based set functions and their approximations.  Each section is not connected to the other sections and some recent papers are missing.
* I think that the equivariant property of set-taking neural networks is more important than the invariant property.  The Deep Sets and Set Transformer papers discussed this topic more thoroughly.  This survey paper also needs to discuss it.
* In Section 1, the authors mention that graph data belongs to set data structures.  I do not agree with this.  Graphs are a generalization of set structures.  Moreover, graph structures cannot be expressed by Definition 2.1.
* I feel like the sentence `Popular conventional neural network architectures like VGG (Simonyan & Zisserman, 2014) and ResNet (He et al., 2016) do not inherently possess the permutation-invariant property` is somewhat outdated in 2024.  I agree that VGG and ResNet are remarkable architectures, but there should be more popular neural networks in 2024.
* In Section 1, the part `The following outlines the organization of this paper` seems redundant.  It is okay to be removed.
* The authors should provide appropriate descriptions on Definitions 2.1, 2.2, 2.3, and 2.4. They are enumerated without any texts.
* Could you provide any references for this sentence `Finally, the output of the network should be invariant to repeated elements, as the presence of duplicate elements should not impact the resulting function approximation`?
* The title of Section 3 should be `Model architectures for approximating set functions`.
* In Section 3.2, Poin tNet should be PointNet.
* How can Set Transformer be expressed by Equation (3)?  It requires more precise expressions.
* In Section 3.4, the authors did not cite the paper of Deep Sets++ and Set Transformer++.
* I cannot find the proper description of Figure 2.  Basically, I think that Figure 2 is redundant in this paper.
* In Figure 3, some taxonomy is not accurate.  For example, a hypergraph is a generalization of a graph.  `Set to hypergraph` should include `Set to graph`.  In addition, why are some topics overlapped?  The authors should describe it more carefully.
* The section title of Section 4.5 should be `Neural processes`.
* I think that the author can detail more thorough future directions of the set-taking neural networks in Section 7.
* Many figures are borrowed from the previous work; I think that it is generally allowed in a survey paper.  However, I believe that they should be redrawn by emphasizing the authors' message on why they are explained in this paper.

---

> ### Author Response · Authors · 2024-02-04
> **Response from the authors.**
>
> Let us thank you for your constructive review, which has helped us to improve the manuscript.
>
> > Connections between sections and lack of recent studies
>
> To improve clarity, we have summarized the contributions and the corresponding sections at the end of Section 1.
>
> In addition, we have added several recent studies (e.g., the additional Subsection 3.9 of Perceiver (Jaegle at el, 2021) and its variants).
>
> > Equivariant property
>
> We have added a new section 2.2, which describes the equivariant property.
>
> > Graph data doesn't belong to set data
>
> We have removed the mentioned description.
>
> > Several neural network architectures are outdated in 2024.
>
> We added more recent architectures: ResNeXt (Xie et al., 2017), EfficientNet (Tan & Le, 2019) and ResNest (Zhang et al., 2022b)
>
> > In Section 1, the part The following outlines the organization of this paper seems redundant
>
> We removed this sentence.
>
> > Appropriate descriptions on Definitions 2.1, 2.2, 2.3, and 2.4.
>
> We have added appropriate explanations before and after the definitions.
>
> > Description without reference
>
> We removed this sentence.
>
> > The title of Section 3
>
> We have corrected the title of section 3 as you indicated .
>
> > Typo in Section 3.2
>
> We fixed typo.
>
> > Set Transformer and Eq.(3)
>
> We have added an additional description of the representation by Eq. (3) of Set Transformer.
>
> > Missing citation of Deep Sets++ and Set Transformer++
>
> We added the citation of Deep Sets++ and Set Transformer++.
>
> > Figure 2 is redundant
>
> We removed this figure.
>
> > Modification of Figure 3
>
> We have corrected the hierarchical relationship between set to graph and hypergraph as you mentioned.
>
> We have also fixed the confusing overlap.
>
> > Section title of Section 4.5
>
> We modified the title of Section 4.5
>
> > Future directions of the set-taking neural networks in Section 7
>
> We have add possible future directions in Section 7.
>
> > Messages for the borrowed figures.
>
> We have added descriptions of why we borrowed these figures.

---

> > ### Comment · Reviewer_Mt8i · 2024-02-05
> >
> > I have read your revision and rebuttal.
> >
> > I think you should say `permutation equivariance` or `permutation equivariant`, instead of `permutation equivalence` or `permutation equivalent`.

---

> > > ### Author Response · Authors · 2024-02-05
> > > **Response from the authors**
> > >
> > > Thanks for pointing that out.
> > > We have corrected the corresponding parts.
> > >
> > > Please see the latest version of our manuscript.

---

### Review · Reviewer_FuAm · 2024-01-22

**Summary Of Contributions:**

This paper provides a survey of various deep learning methods associated with the task of learning permutation invariant functions on sets. The task is motivated by providing applications where learning set functions is more useful than conventional vector-based neural networks.

A broad range of architectures are discussed with their respective use-cases and limitations. The paper also describes theoretical works focused on showing universal approximation properties of popular architectures. Additionally, it also lists datasets that can be useful for training networks on different tasks.

**Audience:**

No

**Claims And Evidence:**

No

**Requested Changes:**

Please refer to my comments in the weaknesses section.

I am willing to improve my recommendation if the authors could provide appropriate additions to the manuscript.

**Strengths And Weaknesses:**

**Strengths:**
- The problem of learning set functions is motivated well and in general the paper is easy to follow.

- As someone from outside this area, I really appreciate the broad overview of the different techniques, tasks and datasets associated with this problem.

**Weaknesses:**
- In my opinion, while the paper describes various useful techniques, it lacks in an effective synthesis of the surveyed works beyond what it is known already. I think the value of the paper can increase significantly if the authors add some generalizable insights on these methods. Some examples could include (but not limited to) shedding light on some key current challenges in the area or perhaps some previously unseen connections between the different approaches that could help with future research.

- Definition 2.2 appears to have a small technical error. Section 2 defines $\mathcal{V}$ as a set of integers whereas $\mathcal{S}$ is a set of d-dimensional vectors $\phi(s_i) \in \mathbb{R}^d$. With that $\mathcal{S}$ should not be a subset of $\mathcal{V}$.

- A very minor issue but I think the writing of the paper is at times a bit verbose and might benefit from some polishing.

---

> ### Author Response · Authors · 2024-02-04
> **Response from the authors.**
>
> We appreciate your very helpful comments.
>
> > Adding some generalizable insights
>
> Based on the survey, we suggested that Deep Sets and its variants can be generalized using a quasi-arithmetic mean, and conducted implementation and experiments in Section 7.
>
> We have also added a future perspectives on the field in Section 8.
>
> > Modification of Definition
>
> To avoid confusion, we have clarified the definitions and the descriptions before and after the definitions.
>
> > Writing of the paper is at times a bit verbose
>
> We have refined the redundant descriptions.

---

### Decision · Action_Editor_MeWm · 2024-03-23

**Recommendation:** Reject

**Comment:**

Even with the revised version, most of reviewers do not want to champion this paper. The paper is a review of existing works on permutation-invariant neural networks, but fails to provide new perspectives or useful insights. The main criticism states that the paper merely lists existing works. In addition, there are multiple ambiguities in the various definitions as well as statements of results. These might lead to confusion.

**Audience:**

This paper is partially useful for audience entering the field of permutation-invariant neural networks, since it provides an overview for a broad range of architectures and approximation results. One benefit of a good survey is that one can spare reading various papers to understand the results. Unfortunately, it is criticized that the current paper does not provide such benefit due to a few reasons that can be found in detailed reviews.

**Claims And Evidence:**

This paper presents a survey of various methods for learning set functions using permutation-invariant neural networks. A broad range of architectures are discussed, providing an extensive list of them. The paper also lists datasets which are commonly used for the evaluation of the models. Lists of various architectures as well as of approximation results are well presented. However, the paper fails to integrate them into a novel relation to another or to give valuable insights.